# Multi-resolution HuBERT: Multi-resolution Speech Self-Supervised Learning with Masked Unit Prediction

**Jiatong Shi**[1]*, **Hirofumi Inaguma**[2], **Xutai Ma**[2], **Ilia Kulikov**[2], **Anna Sun**[2]
[1] Language Technologies Institute, Carnegie Mellon University; [2] Meta AI
`jiatongs@cs.cmu.edu`
`{hirofumii, xutaima, kulikov, annaysun}@meta.com`

## Abstract

Existing Self-Supervised Learning (SSL) models for speech typically process speech signals at a fixed resolution of 20 milliseconds. This approach overlooks the varying informational content present at different resolutions in speech signals. In contrast, this paper aims to incorporate multi-resolution information into speech self-supervised representation learning. We introduce a SSL model that leverages a hierarchical Transformer architecture, complemented by HuBERT-style masked prediction objectives, to process speech at multiple resolutions. Experimental results indicate that the proposed model not only achieves more efficient inference but also exhibits superior or comparable performance to the original HuBERT model over various tasks. Specifically, significant performance improvements over the original HuBERT have been observed in fine-tuning experiments on the LibriSpeech speech recognition benchmark as well as in evaluations using the Speech Universal PERformance Benchmark (SUPERB) and Multilingual SUPERB (ML-SUPERB).

## 1 Introduction

In physics, speech is defined as a vibration that propagates as an acoustic wave through a transmission medium (Fitz, 2007). In the field of speech processing, speech signals are stored using techniques such as sampling and quantization. This results in a discretized abstraction of the original waveform, in both time and amplitude (Roberts & Mullis, 1987).

In practical real-world scenarios, the sampling rate for speech signals can vary between 8 kHz and 48 kHz. High sampling rates can pose challenges for processing due to complications in analyzing long sequences. Typically, speech signals exhibit short-term stationarity within intervals ranging from 10 to 30ms (Zhu & Alwan, 2000). Taking these factors into account, past research has recommended frame-wise processing of speech signals, with frames being extracted over localized sample points (Huang et al., 2001). Traditional spectral feature extraction methods, often based on psychoacoustics, utilize short-term Fourier transform over windows ranging from 20 to 40ms, with shifts between 10 and 30ms (Huang et al., 2001; Davis & Mermelstein, 1980; Hermansky, 1990).

While these conventional spectral features exhibit properties that align well with human psychoacoustics, speech processing systems relying on these features require large volumes of transcribed audio data to achieve high performance (Yu & Deng, 2016). In contrast, Self-Supervised Learning (SSL) speech models utilize unlabeled speech data to generate contextualized speech representations (Oord et al., 2018; Liu et al., 2020a; Baevski et al., 2020; Hsu et al., 2021a; Chung et al., 2021; Chiu et al., 2022; Chen et al., 2022a). These SSL models have shown superior capabilities in contextualizing speech, achieving state-of-the-art results on various benchmarks and challenges (Panayotov et al., 2015; Yang et al., 2021; Evain et al., 2021; Mohamed et al., 2022; Shi et al., 2023a; Agrawal et al., 2023). Moreover, they demonstrate excellent generalizability to low-resource tasks (Baevski et al., 2020; Hsu et al., 2021a; Berrebbi et al., 2022; Zhao & Zhang, 2022). Despite these advancements, existing speech SSL models predominantly follow a similar approach when it comes

---

*The work was conducted by Jiatong Shi during his summer internship at Meta.

to processing speech signals. They typically extract speech frames of 20ms as their fundamental units for pre-training (Baevski et al., 2020; Hsu et al., 2021a; Chung et al., 2021; Chiu et al., 2022; Chen et al., 2022a). This extraction can be accomplished using either a convolutional feature extractor (Baevski et al., 2020; Hsu et al., 2021a; Chen et al., 2022a) or traditional features like Mel filter banks (Lin et al., 2022b; Barrault et al., 2023).

Notably, this uniform frame size of 20ms may not be universally optimal across different downstream tasks. In line with conventional spectral features, existing literature suggests that multi-resolution modeling could enhance performance in various speech processing tasks, such as Automatic Speech Recognition (ASR) (Mallidi & Hermansky, 2016; Mallidi et al., 2018; Hermansky, 2013; Han et al., 2021; Luo et al., 2021; Li et al., 2019b; Andrusenko et al., 2023; Kim et al., 2022; Burchi & Vielzeuf, 2021), Speaker Verification (SV) (Gao et al., 2022), Speech Enhancement (SE) (Zhao et al., 2021; Zhang et al., 2019), and Voice Conversion (VC) (Li et al., 2022). Supporting this notion, recent work by Shi et al. (2023d) demonstrated the advantages of multi-resolution training by using three separate SSL models. Their findings indicate that combining these models focusing on different representations can yield superior results across various tasks, whether used in fine-tuning or as frozen feature extractors. However, the method needs to train different SSL models for each resolution, resulting in a huge computation burden from pre-training.

Despite existing efforts to utilize SSL models for speech at multiple resolutions, no work has explicitly addressed the integration of multi-resolution information during the pre-training phase. This study aims to fill that gap by focusing on multi-resolution pre-training for speech representation. We introduce a novel hierarchical framework, namely multi-resolution HuBERT (MR-HuBERT) designed to encode speech information across multiple resolutions in a single model. The model is pre-trained using objectives for multi-resolution masked unit prediction, which are integrated with HuBERT-style clustering units (Hsu et al., 2021a). Our model shows substantial performance improvements over baseline SSL models across a variety of benchmarks. These include different subsets of the LibriSpeech dataset, the Speech Universal PERformance Benchmark (SUPERB), and the Multilingual SUPERB (ML-SUPERB) (Panayotov et al., 2015; Yang et al., 2021; Shi et al., 2023a). Another of the key advantages of our approach is efficiency; the reduced sequence length resulting from multi-resolution processing enables faster inference to 9-13% computation reduction. We have made the implementation of MR-HuBERT, along with the pre-trained models, available as open-source resources on Fairseq and S3PRL (Ott et al., 2019; Yang et al., 2021).[1]

## 2 BACKGROUND

Self-supervised learning has achieved remarkable success in a wide array of domains, such as computer vision and natural language processing. As detailed in Section 1, similar advancements have been made in the speech processing community. According to the classification scheme by Mohamed et al. (2022), current speech SSL models can be categorized into generative, contrastive, and predictive approaches. Among these, predictive models have shown particularly promising results in recent benchmarks for SSL representation (Yang et al., 2021; Feng et al., 2023; Wang et al., 2021b; Masuyama et al., 2023; Hsu et al., 2021a; Chen et al., 2022a).

As introduced in Section 1, speech SSL models can be applied to various downstream tasks through either fine-tuning or as frozen feature extractors. The architecture of the downstream models can vary widely, including a simple linear probing layer, recurrent neural network layers, Transformer layers, or more complex encoder-decoder frameworks (Baevski et al., 2020; Hsu et al., 2021a; Chung et al., 2021; Yang et al., 2021; Chang et al., 2021; Shi et al., 2023a; Inaguma et al., 2023; Barrault et al., 2023). In all these applications, SSL models generate a sequence of hidden representations with a fixed frameshift, usually around 20ms, which serve as inputs to the downstream tasks.

Two models that have notably excelled in recent benchmarks are HuBERT and WavLM (Hsu et al., 2021a; Chen et al., 2022a). HuBERT employs quantized features for masked unit prediction in the context of masked speech signals (Hsu et al., 2021a). Specifically, the model uses the classic $K$-means algorithm with a fixed cluster size $K$ to perform quantization, where cluster centroid IDs represent the target for each 20ms frame. A noteworthy aspect of HuBERT's pre-training strategy

---

[1]Fairseq: https://github.com/facebookresearch/fairseq/tree/main/examples/mr_hubert; S3PRL: https://s3prl.github.io/s3prl/tutorial/upstream_collection.html#multiresolution-hubert-mr-hubert.

is its iterative training concept. Initially, clustering is performed on Mel Filter-bank Cepstral Coefficients (MFCC), termed as the first iteration. Subsequently, a hidden layer from the first iteration model is extracted and clustered to improve performance. Through this two-stage iterative approach, HuBERT has been shown to either match or exceed the performance of prior state-of-the-art models across various tasks (Hsu et al., 2021a; Yang et al., 2021). With a similar training scheme as HuBERT, WavLM differentiates itself by employing modified self-attention mechanisms and incorporating utterance mixing as a data augmentation technique. As these modifications are not the focus of this paper, our work mainly focuses on the framework of HuBERT and extends over that.

## 3 MR-HUBERT

### 3.1 HUBERT

Consider a sequence of single-channel speech signal $S \in \mathbb{R}^{1 \times L_s}$, where $L_s$ represents the length of the speech signal. For a given iteration $q$, the speech signal $S$ is initially quantized by a pre-trained $K$-means clustering model $g^q(\cdot)$, which is trained on the hidden states from the $q-1$ iteration.[2]

As detailed in Sections 1 and 2, HuBERT employs a convolutional feature extractor $f_0^q(\cdot)$ to first transform the speech signal $S$ into hidden representations at a frame size of 20ms. Following the masking strategies of wav2vec 2.0 and SpanBERT (Baevski et al., 2020; Joshi et al., 2020), $\alpha\%$ of the frames are chosen randomly as starting indices, and $l$ subsequent frames are masked. The set of masked indices is denoted by $\mathbb{M}$.

A Transformer encoder $f_1^q(\cdot)$ is then tasked with predicting the quantized clusters of the masked regions, utilizing cross-entropy loss. The loss function at iteration $q$ is given by:

$$\mathcal{L}_m^q(\theta; S, \mathbb{M}, g^q) = \sum_{t \in \mathbb{M}} \log p_\theta(g^q(S) \mid \tilde{H}_0^q, t), \tag{1}$$

where $\theta$ is the model parameters, $\tilde{H}_0^q$ denotes the masked speech frames from the convolutional feature extractor and $t$ is the time step. It is worth noting that while one could define an unmasked loss $\mathcal{L}_u$, previous experiments have shown that this does not yield significant improvements in the quality of HuBERT's pre-training (Hsu et al., 2021a).

### 3.2 ARCHITECTURE

The proposed architecture for MR-HuBERT is schematically shown in Figure 1. For this explanation, we exemplify a model with two resolutions. This architecture employs a hierarchical Transformer to explicitly encode hidden representations at multiple resolutions while retaining the iterative strategy found in the original HuBERT. The components of the framework are as follows:

Given an speech signal $S$, the convolutional feature extractor $f_0^q$ yields frame-wise feature $H_0 \in \mathbb{R}^{L_{R_1} \times D}$ at a high resolution $R_1$. $L_{R_1}$ is the frame length and $D$ is the feature dimension, which corresponds to the size of the convolutional channels. As outlined in Section 3.1, a masking function $m(\cdot, \mathbb{M})$ is applied to $H_0$ to generate a sequence of masked features $\tilde{H}_0 \in \mathbb{R}^{L_{R_1} \times D}$. This function replaces the feature frames corresponding to the indices in $\mathbb{M}$ with zero vectors.

Next, the masked features $\tilde{H}_0$ are processed by a HuBERT-style Transformer encoder $f_1^q$, noted as High Resolution Transformer Encoder in Figure 1 to produce $\tilde{H}_1^q$. The encoder consists of a pre-convolutional module as well as a stack of transformer layers. The pre-convolutional module includes a 1D-convolutional layer, followed by Layer Normalization and a GELU activation function.

After the high-resolution encoding, the output $\tilde{H}_1^q \in \mathbb{R}^{L_{R_1} \times D}$ is subjected to a downsampling module DOWN$(\cdot)$ to produce a downsampled representation $\tilde{H}_2^q \in \mathbb{R}^{L_{R_2} \times D}$. Here, $R_2$ denotes the lower resolution, and $L_{R_2}$ is the corresponding length of the downsampled hidden representation. The downsampled $\tilde{H}_2^q$ serves as the input for a Low Resolution Transformer Encoder $f_2^q$, as illustrated in Figure 1. Unlike $f_1^q$, $f_2^q$ does not include a pre-convolutional module. Its output $\tilde{H}_3^q$, when coupled with a linear projection, is utilized to predict low-resolution units $g_{R_2}^q(S) \in \mathbb{N}^{+L_{R_2}}$ based

---

[2]The initial iteration ($q = 0$) employs representations derived from MFCC features.

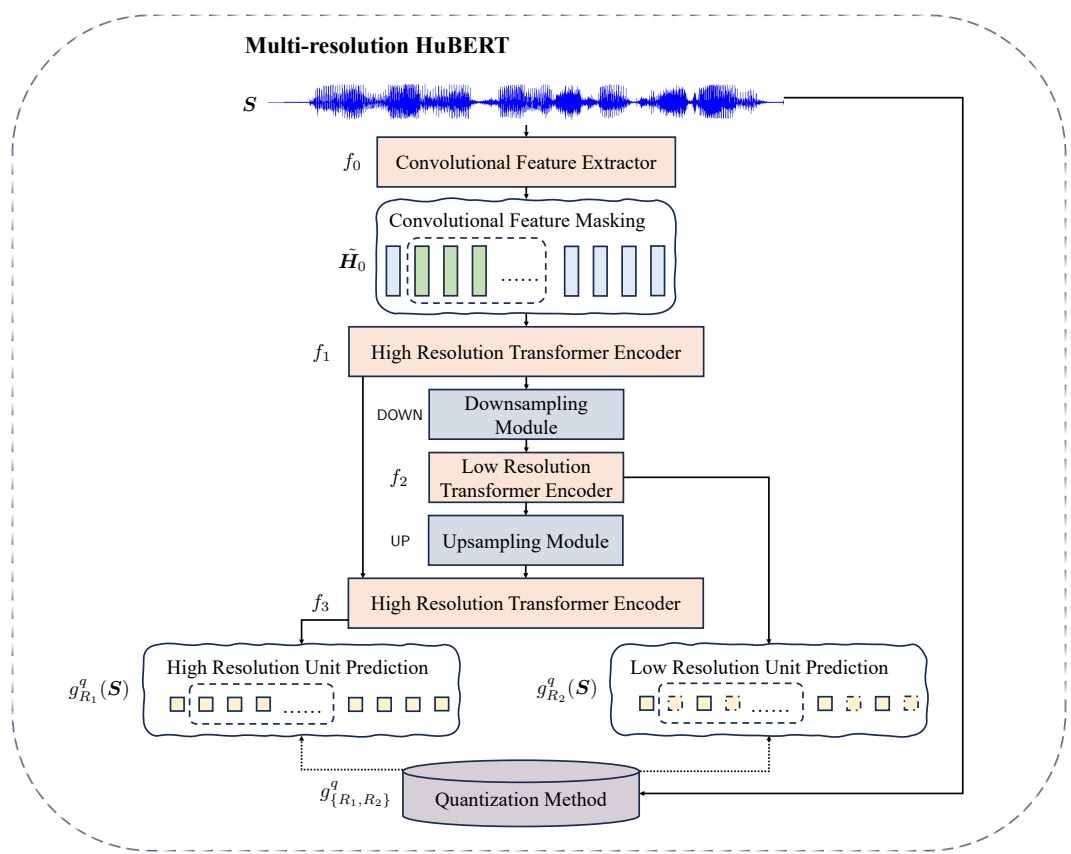

Figure 1: MR-HuBERT pre-training framework. The framework utilizes multi-resolution masked units prediction. The details of each module are discussed in Section 3

on the quantization method $g_{R_2}^q(\cdot)$, detailed in Section 3.4. The whole process of generating $\tilde{\boldsymbol{H}}_3^q$ can be summarized into:

$$\tilde{\boldsymbol{H}}_3^q = f_2^q \circ \mathsf{DOWN} \circ f_1^q(m(f_0^q(\boldsymbol{S}), \mathbb{M})). \tag{2}$$

Finally, an upsampling module $\mathsf{UP}(\cdot)$ expands $\tilde{\boldsymbol{H}}_3^q$ back to high resolution $R_1$, resulting in $\tilde{\boldsymbol{H}}_4^q \in \mathbb{R}^{L_{R_1} \times D}$. This output, when summed with $\tilde{\boldsymbol{H}}_1^q$, is fed into another High Resolution Transformer Encoder $f_3^q(\cdot)$. The ultimate output $\tilde{\boldsymbol{H}}_5^q \in \mathbb{R}^{L_{R_1} \times D}$ is then employed to predict high-resolution units obtained via the quantization method $g_{R_1}^q(\cdot)$. Given $\tilde{\boldsymbol{H}}_3^q$, the process of generating $\tilde{\boldsymbol{H}}_5^q$ can be summarized into:

$$\tilde{\boldsymbol{H}}_5^q = f_3^q(\mathsf{UP}(\tilde{\boldsymbol{H}}_3^q) + f_1^q(m(f_0^q(\boldsymbol{S}), \mathbb{M}))). \tag{3}$$

## 3.3 SAMPLING MODULES

As introduced in Section 3.2, the proposed architecture utilizes an upsampling module $\mathsf{UP}(\cdot)$ and a downsampling module $\mathsf{DOWN}(\cdot)$. The two sampling modules share the same design, as illustrated in Figure 2. The architecture is adapted from the multi-resolution fusion module in Shi et al. (2023d).

To exemplify, we consider the downsampling module. The module first rescale $\tilde{\boldsymbol{H}}_1^q$ into a higher resolution $R_1 \cdot R_1'$ through De-Convolutional Upsampler $\mathrm{DeConv}(\cdot)$ and Repeat-Upsampler $\mathrm{Repeat}(\cdot)$, respectively.[3] The output, $\tilde{\boldsymbol{H}}_1^{q-\mathrm{up}} \in \mathbb{R}^{(L_{R_1} \cdot R_1') \times D}$ is fed into a Convolutional Downsampler $\mathrm{Conv}(\cdot)$ and a Skip-Downsampler $\mathrm{Skip}(\cdot)$, respectively. The final output of the downsampling

---

[3]Given $\tilde{\boldsymbol{H}}_1^q \in \mathbb{R}^{L_{R_1} \times D}$ and the target resolution $R_2$, $R_1'$ and $R_2'$ are the numerator and denominator of the reduced fraction between $R_1$ and $R_2$. They are used as the upsampling factor and the downsampling factor, respectively.

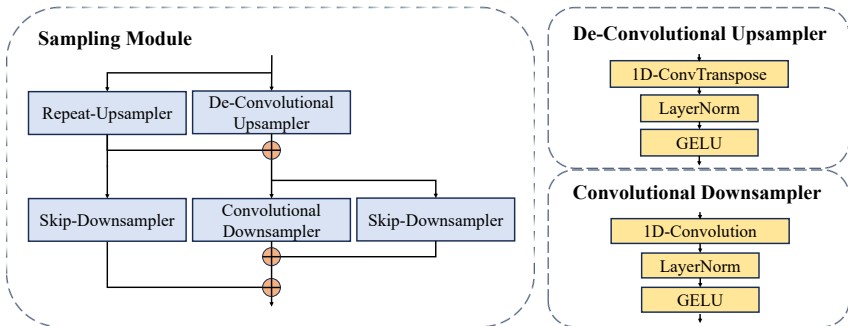

Figure 2: Sampling modules. The proposed sampling modules utilize a residual-based learning framework in either upsampling or downsampling. Details of the module are discussed in Section 3.3.

module, denoted as $\tilde{\boldsymbol{H}}_2^q$ in Section 3.2, is the defined as:

$$\tilde{\boldsymbol{H}}_2^q = \phi \cdot [\mathrm{Skip}(\mathrm{Repeat}(\tilde{\boldsymbol{H}}_1^q)) + \phi \cdot (\mathrm{Conv}(\tilde{\boldsymbol{H}}_1^{q-\mathrm{up}}) + \mathrm{Skip}(\tilde{\boldsymbol{H}}_1^{q-\mathrm{up}}))] \tag{4}$$

### 3.4 OBJECTIVES

Similar to HuBERT discussed in Section 3.1, the objectives of MR-HuBERT focus on masked unit prediction. The major design question for MR-HuBERT, however, is how to construct units for different resolutions. In our experiments discussed in Section 4, we compare different settings in multi-resolution units preparation. The default and most effective approach is simply start from high resolution units extraction and then subsample the low resolution units to match the low resolution sequence from the the Low Resolution Transformer Encoder $f_2^q$. The high resolution units extraction process is similar to HuBERT, by applying $K$-means over hidden representations from $q-1$ iteration. To be specific, $g_{R_1}^q(\cdot)$ is the $K$-means model, where $g_{R_2}^q$ is $g_{R_1}^q \circ d(\cdot)$, where $d$ is a subsampling function.

The pre-training involves two losses: one for high-resolution and another for low-resolution masked unit prediction:

$$\mathcal{L}_m^{q-\{\mathrm{high,\,low}\}}(\theta_{\{\mathrm{high,\,low}\}}; \boldsymbol{S}, \mathbb{M}, g_{\{R_1, R_2\}}^q) = \sum_{t \in \mathbb{M}} \log p_{\theta_{\{\mathrm{high,\,low}\}}}(g_{\{R_1, R_2\}}^q(\boldsymbol{S})|\tilde{\boldsymbol{H}}_0^q, t), \tag{5}$$

where $\theta_{\mathrm{high}}$ are the model parameters of the MR-HuBERT, while $\theta_{\mathrm{low}}$ are partial model parameters that exclude $\mathrm{UP}(\cdot)$ and $f_3^q(\cdot)$. The final objective combines these losses:

$$\mathcal{L}_m^q = \beta \cdot \mathcal{L}_m^{q-\mathrm{high}} + \gamma \cdot \mathcal{L}_m^{q-\mathrm{low}}, \tag{6}$$

where $\beta$ and $\gamma$ are hyperparameters.

## 4 EXPERIMENTS

We evaluate the proposed methods using a variety of speech processing tasks, segmented into four key categories: speech recognition on the LibriSpeech benchmarks (Panayotov et al., 2015), SUPERB benchmark evaluation (Yang et al., 2021) and multilingual SUPERB (ML-SUPERB) benchmark evaluation (Shi et al., 2023a;b).

### 4.1 PRE-TRAINING

**Datasets**: We perform pre-training on three corpora: LibriSpeech (Panayotov et al., 2015), Libri-Light (Kahn et al., 2020), and Voxpopuli (Wang et al., 2021a). LibriSpeech and LibriLight focus exclusively on English, while Voxpopuli is a multilingual dataset encompassing 23 European languages. The total dataset sizes amount to 960 hours for LibriSpeech, 60,000 hours for LibriLight, and 100,000 hours for Voxpopuli.[4]

---

[4]We use the same 100,000 hours split as Wang et al. (2021a).

**Model Configuration**: Following previous work in self-supervised speech learning (Baevski et al., 2020; Hsu et al., 2021a; Chen et al., 2022a), we employ two model sizes for pre-training: *base* and *large*. As outlined in Section 3, we evaluate a two resolution variant of MR-HuBERT with 40ms and the commonly used 20ms. Ablation studies concerning resolutions are elaborated in Appendix B.2.

For both the *base* and *large* models, we adhere to the configurations used in the original HuBERT model (Hsu et al., 2021a). Each encoder (i.e., $f_1^q(\cdot)$, $f_2^q(\cdot)$, and $f_3^q(\cdot)$) as detailed in Section 3.2, has an evenly assigned number of Transformer layers. Specifically, the *base* model uses a four-layer Transformer for each encoder, whereas the *large* model deploys an eight-layer Transformer for each encoder. For an in-depth discussion on the effects of layer allocation, please refer to Appendix B.1.

**Unit Preparation**: To enhance efficiency of pre-training, we directly extract units from the publicly available `HuBERT-base`[5]. We first train a $K$-means model on 50% of the LibriSpeech training set, with $K = 1,000$. Subsequently, the pre-trained $K$-means model is employed to extract target units from LibriSpeech, LibriLight, and Voxpopuli datasets. For multi-resolution scenarios, we perform subsampling of target units by skipping every second unit. Further experiments on unit extraction variants are available in Appendix B.7.

**Pre-trained Models**: We pre-train monolingual and multilingual models for both *base* and *large* settings. Specifically, **mono-base** and **mono-large** are trained on LibriSpeech (960 hours) and LibriLight (60,000 hours) respectively for 400,000 steps. The **multi-base** model is trained on Voxpopuli (384,000 hours) for 800,000 steps. More training details are available in Appendix A.

**Baselines**: Our primary comparisons are made with HuBERT models of matching sizes, specifically `HuBERT-base` and `HuBERT-large`. As noted in the Unit Preparation part, units are consistently extracted from `HuBERT-base`. To account for this, we include an additional iteration trained on this *base* architecture, referred to as `HuBERT-base`+. Furthermore, recognizing that our $K$-means model may not be identical to the one used in `HuBERT-large`, we introduce another setting that uses the same *large* configuration but with our extracted units; we label this as `HuBERT-large`*. For multilingual experiments, we include the public multilingual `mHuBERT-base`, introduced in Lee et al. (2022b) as well as a multilingual `HuBERT-base`* that is trained with the same training configuration of **multi-base**.

To isolate the effects of individual components in our MR-HuBERT, we perform additional ablation studies detailed in Appendix B. These studies encompass mono-resolution models, models using a single high-resolution pre-training target, models with simplified sampling modules, models with less complex settings, etc.

## 4.2 SPEECH RECOGNITION

**Experimental Settings**: We conduct speech recognition experiments using various subsets of the LibriSpeech corpus for training. Specifically, we fine-tune the SSL models as a whole encoder using 1-hour, 10-hour, and 100-hour training subsets. Subsequently, we evaluate each fine-tuned model on four evaluation sets, namely dev-clean, test-clean, dev-other, and test-other. For training configurations, we adhere to the established settings with Connectionist Temporal Classification (CTC) used in wav2vec 2.0 and HuBERT, as outlined in the Fairseq framework (Ott et al., 2019).[6] Beyond decoding via beam search directly from the fine-tuned acoustic model, we also incorporate language model shallow fusion for enhanced performance (Karita et al., 2019). To ensure result reproducibility, we employ an open-source four-gram language model pre-trained on LibriSpeech textual data, along with its associated lexicon (Panayotov et al., 2015).[7] Our chosen evaluation metric is the Word Error Rate (WER).

**Results**: Our findings, illustrated in Table 1, provide compelling evidence of the efficacy of our introduced methods. When subjected to a range of training durations—namely, 1-hour, 10-hour, and 100-hour—the techniques we have implemented consistently surpass the Word Error Rate (WER) results of the four reference baseline models. In the *base* model variant, the **mono-base** model we introduce consistently showcases a marked 1%-2% WER improvement across the board, when measured against all four evaluation datasets. For the *large* model configuration, the results become even

---

[5]https://dl.fbaipublicfiles.com/hubert/hubert_base_ls960.pt

[6]https://github.com/facebookresearch/fairseq

[7]https://www.openslr.org/11/

Table 1: Word error rate for speech recognition on LibriSpeech benchmark, evaluated on 1-hour, 10-hour and 100-hour labeled data. Results with a 4-gram language model joint decoding are in parentheses. Model settings are discussed in Section 4.1.

| Model | Unlabeled Data (h) | dev-clean | dev-other | test-clean | test-other |
|---|---|---|---|---|---|
| | | *1-hour labeled* | | | |
| `HuBERT-base` | 960 | 20.17 (8.75) | 28.11 (16.09) | 20.64 (8.88) | 28.87 (16.71) |
| `HuBERT-base`[+] | 960 | 19.64 (8.14) | 25.08 (12.36) | 20.15 (8.31) | 25.63 (12.82) |
| `HuBERT-large` | 60,000 | 14.42 (5.84) | 18.80 (9.53) | 14.40 (5.81) | 19.29 (9.91) |
| `HuBERT-large`[*] | 60,000 | 15.09 (4.30) | 18.20 (**6.84**) | 14.90 (4.30) | 18.05 (**7.23**) |
| **`mono-base`** | 960 | 18.78 (7.33) | 23.72 (11.53) | 19.26 (7.41) | 24.46 (12.14) |
| **`mono-large`** | 60,000 | **6.44 (3.64)** | **10.94** (6.85) | **6.37 (3.75)** | **11.41 (7.23)** |
| | | *10-hour labeled* | | | |
| `HuBERT-base` | 960 | 9.62 (4.88) | 16.60 (8.51) | 9.71 (4.97) | 17.00 (9.15) |
| `HuBERT-base`[+] | 960 | 9.51 (4.85) | 14.27 (8.37) | 9.72 (4.88) | 14.89 (8.94) |
| `HuBERT-large` | 60,000 | 5.68 (3.27) | 8.67 (5.51) | 5.75 (3.50) | 8.96 (5.93) |
| `HuBERT-large`[*] | 60,000 | 5.61 (3.24) | 8.68 (5.55) | 5.57 (3.25) | 9.02 (6.00) |
| **`mono-base`** | 960 | 8.51 (4.80) | 13.18 (8.29) | 8.46 (4.91) | 13.51 (8.33) |
| **`mono-large`** | 60,000 | **5.58 (3.12)** | **8.57 (5.44)** | **5.52 (3.15)** | **8.74 (5.86)** |
| | | *100-hour labeled* | | | |
| `HuBERT-base` | 960 | 5.76 (3.66) | 12.90 (8.45) | 5.81 (3.84) | 12.76 (8.48) |
| `HuBERT-base`[+] | 960 | 5.71 (3.33) | 10.66 (6.51) | 5.97 (3.55) | 10.87 (7.09) |
| `HuBERT-large` | 60,000 | 3.11 (2.37) | **6.01 (4.22)** | 3.14 (2.48) | 6.15 (4.67) |
| `HuBERT-large`[*] | 60,000 | **3.03** (2.44) | 6.30 (4.61) | 3.12 (2.62) | 6.14 (4.69) |
| **`mono-base`** | 960 | 4.89 (3.21) | 9.04 (6.47) | 4.92 (3.57) | 9.17 (6.81) |
| **`mono-large`** | 60,000 | 3.06 (**2.33**) | 6.04 (4.54) | **3.01 (2.44)** | **5.98 (4.61)** |

more compelling. The **`mono-large`** model, in particular, stands out: when trained on the 1-hour dataset, it achieves a WER reduction oscillating between 40% and 50%. For the 10-hour training set, the dev-other and test-other evaluation datasets reflect the most pronounced improvements. Shifting to the 100-hour training set, the test-clean and test-other sets emerge as the beneficiaries of the largest boosts in performance. Furthermore, when a joint-decoding strategy with the language model is in place, while the performance differential becomes less pronounced, the proposed MR-HuBERT still maintains a performance edge, always matching or outperforming the baseline HuBERT models. A salient takeaway is that our proposed models consistently rival or outstrip the baseline models, underscoring the robustness and superiority of the methodologies we've employed.

## 4.3 SUPERB EVALUATION

**Experimental Settings**: Our evaluation within the SUPERB framework aims to provide a holistic assessment of the quality of SSL representations across a broad array of speech processing tasks (Yang et al., 2021; Tsai et al., 2022; Feng et al., 2023). Specifically, we assess our proposed models on tasks including Phone Recognition (PR), Automatic Speech Recognition (ASR), Intent Classification (IC), Keyword Spotting (KS), Slot Filling (SF), Speech Translation (ST), Speech Enhancement (SE), and Speech Separation (SS).[8]

To ensure consistent evaluations, we adopt metrics outlined in Yang et al. (2021): Phone Error Rate (PER) for PR, WER for ASR, Accuracy (ACC) for IC and KS, F-1 measure and Character Error Rate (CER) for SF, BLEU for ST, Short-Time Objective Intelligibility (STOI) and Perceptual Evaluation of Speech Quality (PESQ) for SE, and Scale-Invariant Signal-to-Distortion Ratio improvement (SI-SDRi) for SS.

Table 2: Categorical SUPERB score. Category information and SUPERB score definition are discussed in Section 4.3.

| Model | Understanding | Enhancement | General |
|---|---|---|---|
| `HuBERT-base` | 861.2 | 98.20 | 670.4 |
| `HuBERT-base`[+] | 876.9 | 150.2 | 695.2 |
| `HuBERT-large` | 932.6 | 456.0 | 813.4 |
| `HuBERT-large`[*] | 936.2 | 501.5 | 827.5 |
| **`mono-base`** | 885.8 | 195.0 | 708.7 |
| **`mono-large`** | **949.7** | **609.5** | **864.6** |

We adhere to the SUPERB policy for downstream model training. In particular, we keep the SSL upstream models fixed and only ad-

---

[8]Besides the SUPERB public benchmark tasks, we also explore Voice Conversion (VC) as outlined in Huang et al. (2022a;b). For more details, see Appendix D.

Table 3: Detailed SUPERB evaluation. Detailed metrics and settings are detailed in Section 4.3.

| Model | | | Understanding | | | | | Enhancement | | |
|---|---|---|---|---|---|---|---|---|---|---|
| | PR($\downarrow$) | ASR($\downarrow$) | IC($\uparrow$) | KS($\uparrow$) | SF-F1($\uparrow$) | SF-CER($\downarrow$) | ST($\uparrow$) | SE-STOI($\uparrow$) | SE-PESQ($\uparrow$) | SS($\uparrow$) |
| HuBERT-base | 5.40 | 6.42 | 98.34 | 96.30 | 88.53 | 25.20 | 15.53 | **0.94** | 2.58 | 9.36 |
| HuBERT-base$^+$ | 4.56 | 6.34 | 98.39 | 96.46 | 89.12 | 23.10 | 16.33 | 0.93 | 2.55 | 9.72 |
| HuBERT-large | 3.54 | 3.62 | 98.76 | 95.29 | 89.81 | 21.76 | 20.01 | **0.94** | 2.64 | 10.45 |
| HuBERT-large$^*$ | 3.59 | **3.53** | 98.73 | 97.70 | 89.88 | 22.51 | 20.02 | **0.94** | 2.65 | 10.61 |
| **mono-base** | 4.16 | 5.76 | 98.68 | 96.49 | 88.96 | 23.59 | 16.94 | **0.94** | 2.55 | 9.92 |
| **mono-large** | **3.15** | 3.78 | **98.76** | **97.76** | **90.57** | **20.60** | **21.05** | **0.94** | **2.67** | **10.97** |

Table 4: Results on ML-SUPERB {10-minute/1-hour} settings. Detailed metrics and settings are detailed in Section 4.4.

| SSL | Monolingual ASR | Multilingual ASR | | LID | Multilingual ASR + LID | | | SUPERB$_s$($\uparrow$) |
|---|---|---|---|---|---|---|---|---|
| | | Normal | Few-shot | Normal | Normal | | Few-shot | |
| | CER/PER($\downarrow$) | CER($\downarrow$) | CER($\downarrow$) | ACC($\uparrow$) | ACC($\uparrow$) | CER($\downarrow$) | CER($\downarrow$) | |
| HuBERT-base | 42.8 / 35.3 | 39.8 / 31.4 | 44.5 / 42.7 | 61.2 / 86.1 | **71.5** / 86.0 | 39.2 / 30.9 | 43.8 / 41.8 | 831.9 / 884.9 |
| HuBERT-base$^+$ | 42.9 / 35.3 | 41.5 / 31.2 | 45.8 / 42.8 | 63.8 / 81.9 | 70.1 / 85.8 | 39.6 / 31.3 | 44.6 / 40.7 | 819.1 / 875.8 |
| HuBERT-large | **38.2** / 32.2 | 44.4 / 37.7 | 48.2 / 43.5 | 46.5 / 64.1 | 55.4 / 77.7 | 45.6 / 35.1 | 49.3 / 42.2 | 678.7 / 783.6 |
| HuBERT-large$^*$ | 41.2 / 32.6 | 42.8 / 32.8 | 45.6 / 42.5 | 42.3 / 58.9 | 59.2 / 84.7 | 42.3 / 29.8 | 44.1 / 41.4 | 704.5 / 817.6 |
| mHuBERT-base | 41.0 / 33.0 | 40.5 / 33.4 | 45.6 / 43.6 | 52.4 / 72.5 | 46.6 / 70.9 | 36.8 / 29.7 | 44.2 / 43.1 | 746.2 / 812.7 |
| mHuBERT-base$^*$ | 40.1 / 32.3 | 36.3 / **27.3** | **38.6** / 39.0 | **64.0** / 82.0 | 70.4 / 84.6 | 35.4 / **27.1** | **39.0** / 37.0 | 950.8 / 964.5 |
| **mono-base** | 42.8 / 34.6 | 40.2 / 30.6 | 45.0 / 42.2 | 67.2 / **86.3** | 68.7 / **86.9** | 40.3 / 30.6 | 44.1 / 41.6 | 843.5 / 899.9 |
| **mono-large** | 40.5 / 32.0 | 38.9 / 29.4 | 42.7 / 40.5 | 45.1 / 75.4 | 67.6 / 85.9 | 39.0 / 29.7 | 43.8 / 40.8 | 785.2 / 905.4 |
| **multi-base** | 38.3 / **30.6** | **34.1** / 27.5 | 39.6 / **38.9** | **64.0** / 85.1 | 69.9 / 84.4 | **34.4** / 28.0 | 40.9 / **36.6** | **957.2** / **986.8** |

just the learning rate. To address reproducibility, we perform a simple grid search for the learning rate, considering only the default rate in S3PRL along with its 0.1x and 10x variations. We also use the weighted summation strategy for the frozen SSL representation. To mitigate the resolution differences across layers, we conduct simple repeat upsampling or skip downsampling as outlined in (Shi et al., 2023d).

To gauge the performance of SSL representations across tasks, we categorize SUPERB tasks into two main clusters: Understanding and Enhancement (Generation). We calculate the SUPERB score (denoted as SUPERB$_s$), as defined in the SLT 2022 SUPERB challenge (Feng et al., 2023), which employs linear scaling between conventional spectral features and state-of-the-art upstream representations in the corresponding tasks. Comprehensive performance metrics that take into account all evaluated tasks are also calculated. More information on the SUPERB is available in Appendix D.

**Results**: The comprehensive results, divided by task category, are presented in Table 2 and Table 3. Our proposed MR-HuBERT demonstrates marked improvements over a variety of understanding and enhancement tasks in both *base* and *large* configurations.

## 4.4 ML-SUPERB EVALUATION

**Experimental Settings**: We evaluate the performance of our proposed multilingual speech processing method using the ML-SUPERB benchmark (Shi et al., 2023a). This benchmark, which is supported by 143 languages, has been implemented as a recipe within the ESPnet framework (Watanabe et al., 2018)[9]. The ML-SUPERB benchmark comprises two sets of general benchmarks—specifically, a 10-minute set and a 1-hour set—across four tasks: Monolingual ASR, Multilingual ASR, Language Identification (LID), and a joint task of Multilingual ASR+LID. To maintain the integrity of the experimental comparison, we adhere to the ML-SUPERB guidelines for downstream architectures and training configurations, including the use of frozen SSL representations (Shi et al., 2023a). For the evaluation, we employ the standard metrics: Character Error Rate (CER) or PER for ASR tasks, and ACC for LID tasks. Furthermore, we calculate a composite ML-SUPERB score as defined by Shi et al. (2023a) to provide an overall measure of performance. Additional information on the SUPERB evaluation is available in Appendix E.

**Results**: Our evaluations on the ML-SUPERB benchmark are summarized in Table 4. The data reveals that our proposed multilingual model, **multi-base**, stands out with the topmost per-

---

[9] https://github.com/espnet/espnet/tree/master/egs2/ml_superb/asr1

formance. Notably, even our monolingual pre-trained models, **mono-base** and **mono-large**, surpass the overall monolingual baselines. Furthermore, they outperform the multilingual model `mHuBERT-base` and `mHuBERT-base`[*] in the overall ML-SUPERB score.

## 4.5 DISCUSSION: INFERENCE SPEED

In addition to achieving notable gains in performance across various test scenarios, the proposed method also offers advantages in terms of computational efficiency, particularly during the inference stage. This efficiency is primarily attributable to the reduced sequence length required for self-attention computations. To quantitatively evaluate this improvement, we employ Multiply-Add Cumulations (MACs) as our metric of comparison between the baseline models and our proposed method. We utilize the TorchProfile toolkit to calculate MACs[10]. Specifically, we analyze audio samples of varying lengths—2s, 4s, 8s, 16s, and 32s—to calculate the total MACs for each method. The results indicate a clear computational advantage for the proposed method: in the *base* model configuration, the total MACs were reduced from 431G to 394G, representing an improvement of 9%. In the *large* model configuration, the MACs decreased from 1116G to 971G, corresponding to a 13% improvement.

## 5 RELATION TO SIMILAR APPROACHES IN OTHER CONTEXTS

The idea of leveraging multiple resolutions has been explored in various other contexts. In speech understanding, downsampled spoken feature sequences are commonly employed to extract high-level linguistic or semantic features for efficiency (Chen et al., 2019; Meng et al., 2023; Chen et al., 2023a) or to better integrate pre-trained language models (Gaido et al., 2021; Shi et al., 2023c; Wu et al., 2023; Li et al., 2023c). In speech synthesis, multi-resolution discriminators have been instrumental in recent adversarial-based vocoders (Yamamoto et al., 2020; Kong et al., 2020; Yoneyama et al., 2023). Additionally, multi-resolution or multi-scale networks have shown robust performance in speech enhancement (Zhang & Wang, 2020; Zhang et al., 2022b; Xiang et al., 2021; Xu et al., 2020; Shi et al., 2019). While prior work exists, our paper stands out for its focus on a novel hierarchical architecture for speech pre-training. The resulting models offer not only substantial performance gains across downstream tasks but also computational efficiencies during inference.

Similar multi-resolution strategies have also found applications in other domains. In computer vision, multi-scale convolutional networks are employed for various tasks such as object detection and human pose estimation (Yang & Ramanan, 2015; Cai et al., 2016; Ghiasi et al., 2019; Mathieu et al., 2016). Among these, Hourglass networks stand out for their hierarchical multi-resolution processing, which has resulted in significant performance gains (Newell et al., 2016; Melekhov et al., 2017; Yang et al., 2017). This concept has been extended to the text domain as the Hourglass transformer, which has proven effective for sequence processing (Zhai et al., 2023; Guo et al., 2022; Nawrot et al., 2023; 2022). Our work has a similar architecture to the Hourglass transformer in speech pre-training with specific features like masked unit prediction, multi-resolution targets, and other speech-related architectural nuances.

## 6 CONCLUSION

This paper introduces MR-HuBERT, a self-supervised speech learning model that extends HuBERT by employing multi-resolution masked unit prediction in conjunction with a hierarchical transformer architecture. Comprehensive evaluations across various benchmarks reveal that MR-HuBERT substantially outperforms the original HuBERT model across a broad spectrum of speech processing tasks. These include, but are not limited to, speech recognition, spoken language understanding, multilingual speech recognition, and speech enhancement. Beyond these performance gains, the model also exhibits computational efficiencies, specifically a 9-13% reduction in computational complexity, addressing efficiency concerns.[11]

---

[10]`https://github.com/zhijian-liu/torchprofile`

[11]Limitations of the work are discussed in Appendix F, while some future directions are discussed in Appendix G.

## 7 ETHICS STATEMENT

The development and implementation of MR-HuBERT represent a significant step forward in self-supervised pre-training for speech models. While this model demonstrates substantial potential and effectiveness across various tasks, it's crucial to approach its adoption and application ethically:

- Openness and Transparency: We remain committed to the principles of open research. By releasing the complete codebase and associated checkpoints of our MR-HuBERT model, we aim to foster an environment of transparency and reproducibility. This initiative encourages peer reviews and allows researchers to independently validate our findings.
- Potential Misuse: Like any advanced technology, MR-HuBERT's capabilities could be misappropriated for malicious purposes. While the model offers enhanced performance across various speech tasks, users must employ it responsibly, respecting individual privacy and avoiding potential misuse in surveillance or unauthorized information extraction. MR-HuBERT presents an unforeseen avenue for speech disentanglement, especially in its large configurations, as detailed in Appendix D. As the model evolves, ensuring that it doesn't unintentionally disentangle or misinterpret cultural nuances, accents, or dialects becomes paramount. This concern is essential for avoiding potential biases or misrepresentations.

While MR-HuBERT represents a promising stride in speech model advancement, its ethical implications are at the forefront of our considerations. We urge the community to employ this technology with caution, respect, and a commitment to the broader good.

## 8 REPRODUCIBILITY STATEMENT

In the spirit of open research and fostering further advancements in the field, we will be releasing the complete codebase associated with our MR-HuBERT model. This encompasses the entire spectrum of models discussed in our work, including models presented in Appendices. Researchers, academicians, and enthusiasts can access, reproduce, and potentially build upon our findings. We believe that this transparent sharing will not only validate our findings but also inspire innovative research directions anchored around MR-HuBERT. Details regarding access and implementation will be updated after the double-blind review. We eagerly anticipate the community's engagement and are open to collaborations, feedback, and further enhancements to the model.

## 9 ACKNOWLEDGEMENT

We extend our heartfelt gratitude to Juan Pino, Paden Tomasello, Changhan Wang, Andy Chung, Ning Dong, Hongyu Gong, and Maha Elbayad for their invaluable advice and unwavering support throughout this project. Their insights and expertise have been indispensable to this work.

Special recognition is owed to Yun Tang and Shinji Watanabe. Their contributions, particularly in the formative stages of our research, have been instrumental. Their guidance in shaping our initial research idea has set a strong foundation for the entirety of this work.

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

Table 5: Detailed Hyper-parameters for models presented in main content.

| | | Additional Baseline | | | Monolingual Models | | Multilingual Models |
| | | HuBERT-base[+] | HuBERT-large[*] | mHuBERT-base[*] | mono-base | mono-large | multi-base |
|---|---|---|---|---|---|---|---|
| Architecture | Num. Param (M) | 95 | 317 | 95 | 97 | 321 | 97 |
| | Transformer Layers | 12 | 24 | 12 | 4 * 3 | 8 * 3 | 4 * 3 |
| | - Attention Dim. | 768 | 1024 | 768 | 768 | 1024 | 768 |
| | - Linear Dim. | 3072 | 4096 | 3072 | 3072 | 4096 | 3072 |
| | - Attention Head | 12 | 16 | 12 | 12 | 16 | 12 |
| | Sampling Module | - | - | - | up+down | up+down | up+down |
| | - Kernel size | - | - | - | 1 | 1 | 1 |
| | - Channel Size | - | - | - | 768 | 1024 | 768 |
| | Conv. Extractor | [(512, 10, 5), (512, 3, 2) * 4, (512, 2, 2) * 2] | | | | | |
| | Mask Ratio | 0.8 | | | | | |
| Training | Num. GPU | 32 | 128 | 32 | 32 | 128 | 32 |
| | Num. Frames | 100k | 90k | 140k | 100k | 30k | 100k |
| | Grad. Accum. | 1 | 1 | 1 | 1 | 3 | 1 |
| | Num. Steps | 400k | 400k | 800k | 400k | 400k | 800k |
| | Optimizer | Adamw | Adamw | Adamw | Adamw | Adamw | Adamw |
| | Learning Rate | 0.0005 | 0.0015 | 0.0005 | 0.0005 | 0.0015 | 0.0005 |
| | Warmup Steps | 32k | 32k | 32k | 32k | 32k | 32k |
| | Dropout | 0.1 | 0.0 | 0.1 | 0.1 | 0.0 | 0.1 |
| | Loss Weights ($\beta, \gamma$) | - | - | - | (1, 1) | (1, 1) | (1, 1) |
| | Audio Norm | true | false | true | true | false | true |

## A PRE-TRAINING SETTINGS

The pre-training configurations of the models presented in the main content can be found in Table 5. Generally, MR-HuBERT possesses a parameter count analogous to the original HuBERT model. We've made concerted efforts to mitigate the impact of incorporating an additional sampling module, which naturally adds more parameters. Specifically, we consistently employ a kernel size of 1 for both convolutional and de-convolutional layers in the sampling module, as elaborated in Section 3.3. Nonetheless, the model experiences a modest increase in parameter size, but this surge is less than 3%. To ensure that the performance boosts highlighted in Section 4 aren't merely due to this increase, we've carried out comprehensive ablation studies, detailed in Appendix B.

In line with the insights from Hsu et al. (2021a), a more substantial batch size can typically augment model performance. In our research, when juxtaposing our method against the baselines, we've meticulously ensured that the batch size of our approach is either equivalent to or smaller than that of the baseline, to offset potential biases. All model training was executed on V100-32GB GPUs using the Fariseq toolkit (Ott et al., 2019).

## B ABLATION STUDIES

To garner an in-depth understanding of MR-HuBERT, we undertake extensive ablation studies. This ensures each component of MR-HuBERT is optimized and offers insight into their individual contributions to the model's superior performance. We delved into seven distinct conditions:

- Encoder Layer Sizes: We explore the effect of varying the layer sizes for each encoder (Appendix B.1).

- Multi-Resolution Analysis: We evaluate the impact of utilizing multiple resolutions (Appendix B.2).

- Simpler Upsampling & Downsampling Modules: A study into the implications of adopting a simplified upsampling or downsampling module is presented (Appendix B.3).

- Single Prediction Target: Instead of multi-tasking, we scrutinize the outcome of using a singular prediction target (Appendix B.4).

- Single Resolution: The performance implications of deploying only one resolution are analyzed (Appendix B.5).

- Compact Model: We test the efficacy of the model in a more compact setting (Appendix B.6).

- Target Units for Prediction: We investigate the repercussions of utilizing various target units for prediction (Appendix B.7).

Table 6: Ablation study configurations on different encoder layer sizes in the *base* setting.

| Model | Layers | Num. Param (M) | MACs (G) |
|---|---|---|---|
| HuBERT-base | 12 | 95 | 431 |
| HuBERT-base[+] | 12 | 95 | 431 |
| **mono-base** | (4, 4, 4) | 97 | 394 |
| (B.1)-a | (2, 4, 6) | 97 | 394 |
| (B.1)-b | (5, 2, 5) | 97 | 416 |
| (B.1)-c | (6, 4, 2) | 97 | 394 |

Table 7: Ablation study of differing encoder layer sizes for the *base* setting. The experiments are conducted on ASR fine-tuning experiments over LibriSpeech subsets.

| Model | Layers | dev-clean | dev-other | test-clean | test-other |
|---|---|---|---|---|---|
| | | *1-hour labeled* | | | |
| HuBERT-base | 12 | 20.17 | 28.11 | 20.64 | 28.87 |
| HuBERT-base[+] | 12 | 19.64 | 25.08 | 20.15 | 25.63 |
| **mono-base** | (4, 4, 4) | 18.78 | 23.72 | 19.26 | 24.46 |
| (B.1)-a | (2, 4, 6) | 18.71 | 23.30 | 19.30 | 23.94 |
| (B.1)-b | (5, 2, 5) | 18.61 | **23.22** | **18.63** | **23.75** |
| (B.1)-c | (6, 4, 2) | **18.41** | 23.37 | 18.83 | 23.96 |
| | | *10-hour labeled* | | | |
| HuBERT-base | 12 | 9.62 | 16.60 | 9.71 | 17.00 |
| HuBERT-base[+] | 12 | 9.51 | 14.27 | 9.72 | 14.89 |
| **mono-base** | (4, 4, 4) | 8.51 | 13.18 | 8.46 | 13.51 |
| (B.1)-a | (2, 4, 6) | 8.61 | 13.33 | 8.54 | 13.64 |
| (B.1)-b | (5, 2, 5) | **8.30** | **12.96** | **8.38** | **13.42** |
| (B.1)-c | (6, 4, 2) | 8.71 | 13.24 | 8.71 | 13.72 |
| | | *100-hour labeled* | | | |
| HuBERT-base | 12 | 5.76 | 12.90 | 5.81 | 12.76 |
| HuBERT-base[+] | 12 | 5.71 | 10.66 | 5.97 | 10.87 |
| **mono-base** | (4, 4, 4) | 4.89 | **9.04** | 4.92 | **9.17** |
| (B.1)-a | (2, 4, 6) | 4.96 | 9.40 | 5.00 | 9.76 |
| (B.1)-b | (5, 2, 5) | **4.65** | 9.22 | **4.78** | 9.44 |
| (B.1)-c | (6, 4, 2) | 5.11 | 9.80 | 5.10 | 9.90 |

The above ablations are all conducted in *base* setting for efficiency, while we also conduct selected *large* setting experiments in Appendix B.8.

As detailed in Section 4.2, we utilize the labeled LibriSpeech subsets of 1-hour, 10-hour, and 100-hour, as described in Kahn et al. (2020), for fine-tuning. The LibriSpeech evaluation sets serve as our testing grounds. All ASR results are presented using the word error rate. Prioritizing the quality of representation, we opt for Viterbi decoding over language model joint decoding. In addition to the ASR performance, we provide information on each model's parameter size and MACs. The calculation of MACs can be found in Section 4.5.

## B.1 ENCODER LAYER SIZES

As discussed in Section 4.1, each encoder of MR-HuBERT maintains a consistent layer size. However, the impact of varied layer sizes for each encoder on the model's efficacy remains an open question. To address this, we explore the *base* setting by altering layer counts.

Table 8: Ablation study configurations on three-resolution MR-HuBERT in the *base* setting.

| Model | Resolutions (ms) | Layers | Num. Param (M) | MACs (G) |
|---|---|---|---|---|
| `HuBERT-base` | 20 | 12 | 95 | 431 |
| `HuBERT-base`[+] | 20 | 12 | 95 | 431 |
| **`mono-base`** | (20, 40) | (4, 4, 4) | 97 | 393 |
| `(B.2)-a` | (20, 40, 80) | (3, 2, 2, 2, 3) | 100 | 353 |
| `(B.2)-b` | (20, 40, 80) | (2, 2, 4, 2, 2) | 100 | 331 |
| `(B.2)-c` | (20, 40, 100) | (2, 2, 2, 2, 2) | 86 | 316 |

Table 9: Ablation study of three-resolution MR-HuBERT in the *base* setting. The experiments are conducted on ASR fine-tuning experiments over LibriSpeech subsets.

| Model | Resolutions (ms) | dev-clean | dev-other | test-clean | test-other |
|---|---|---|---|---|---|
| | | *1-hour labeled* | | | |
| `HuBERT-base` | 20 | 20.17 | 28.11 | 20.64 | 28.87 |
| `HuBERT-base`[+] | 20 | 19.64 | 25.08 | 20.15 | 25.63 |
| **`mono-base`** | (20, 40) | **18.78** | **23.72** | **19.26** | **24.46** |
| `(B.2)-a` | (20, 40, 80) | 19.63 | 24.60 | 19.80 | 24.93 |
| `(B.2)-b` | (20, 40, 80) | 19.93 | 24.08 | 19.79 | 25.32 |
| `(B.2)-c` | (20, 40, 100) | 19.11 | 24.76 | 19.48 | 25.00 |
| | | *10-hour labeled* | | | |
| `HuBERT-base` | 20 | 9.62 | 16.60 | 9.71 | 17.00 |
| `HuBERT-base`[+] | 20 | 9.51 | 14.27 | 9.72 | 14.89 |
| **`mono-base`** | (20, 40) | **8.51** | **13.18** | **8.46** | **13.51** |
| `(B.2)-a` | (20, 40, 80) | 8.63 | 14.19 | 8.84 | 14.31 |
| `(B.2)-b` | (20, 40, 80) | 8.81 | 14.34 | 8.90 | 14.61 |
| `(B.2)-c` | (20, 40, 100) | 9.34 | 15.08 | 9.48 | 15.15 |
| | | *100-hour labeled* | | | |
| `HuBERT-base` | 20 | 5.76 | 12.90 | 5.81 | 12.76 |
| `HuBERT-base`[+] | 20 | 5.71 | 10.66 | 5.97 | 10.87 |
| **`mono-base`** | (20, 40) | 4.89 | **9.04** | 4.92 | **9.17** |
| `(B.2)-a` | (20, 40, 80) | **4.70** | 10.04 | **4.87** | 9.90 |
| `(B.2)-b` | (20, 40, 80) | 5.00 | 10.49 | 5.10 | 10.37 |
| `(B.2)-c` | (20, 40, 100) | 5.53 | 11.47 | 5.60 | 11.25 |

The model configurations for this exploration are detailed in Table 6. Across all new configurations, the parameter size remains consistent. Yet, in the `(B.1)-b` configuration, where low-resolution layers are minimized, the MACs rise to 416G from 394G.

The evaluation outcomes are tabulated in Table 7. A key insight drawn from these results is that the `(B.1)-b` configuration excels in most LibriSpeech evaluation scenarios, especially when working with limited labeled data sets like the 1-hour and 10-hour subsets. This underscores the notion that while low-resolution modeling can effectively learn with fewer layers, the contribution of high-resolution comprehension remains pivotal to the overall model's success.

## B.2 MULTI-RESOLUTION ANALYSIS

While the main discussion primarily revolves around MR-HuBERT trained with two resolutions, this section explores its performance using three resolutions. This is to gauge the potential advantages or drawbacks of adopting more than two resolutions. Table 8 showcases that by adding a lower resolution, there's an increase in the parameter size to 100M, primarily due to the inclusion of extra

Table 10: Ablation study on simplified upsampling & downsampling modules along with a singular prediction target in the *base* setting. The experiments are conducted on ASR fine-tuning experiments over LibriSpeech subsets.

| Model | Note | dev-clean | dev-other | test-clean | test-other |
|---|---|---|---|---|---|
| *1-hour labeled* | | | | | |
| `HuBERT-base` | - | 20.17 | 28.11 | 20.64 | 28.87 |
| `HuBERT-base`[+] | - | 19.64 | 25.08 | 20.15 | 25.63 |
| **`mono-base`** | - | 18.78 | 23.72 | 19.26 | 24.46 |
| `(B.3)-a` | Simple sampling | **18.06** | **22.61** | **18.33** | **23.37** |
| `(B.4)-a` | Single target | 19.74 | 25.12 | 20.04 | 25.87 |
| `(B.4)-b` | Simple sampling + Single target | 19.02 | 24.30 | 19.40 | 24.94 |
| *10-hour labeled* | | | | | |
| `HuBERT-base` | - | 9.62 | 16.60 | 9.71 | 17.00 |
| `HuBERT-base`[+] | - | 9.51 | 14.27 | 9.72 | 14.89 |
| **`mono-base`** | - | 8.51 | 13.18 | **8.46** | 13.51 |
| `(B.3)-a` | Simple sampling | **8.30** | **12.88** | 8.49 | **13.35** |
| `(B.4)-a` | Single target | 9.43 | 14.49 | 9.52 | 14.99 |
| `(B.4)-b` | Simple sampling + Single target | 9.15 | 13.78 | 9.22 | 14.42 |
| *100-hour labeled* | | | | | |
| `HuBERT-base` | - | 5.76 | 12.90 | 5.81 | 12.76 |
| `HuBERT-base`[+] | - | 5.71 | 10.66 | 5.97 | 10.87 |
| **`mono-base`** | - | **4.89** | **9.04** | **4.92** | **9.17** |
| `(B.3)-a` | Simple sampling | 4.91 | 9.66 | 5.10 | 9.73 |
| `(B.4)-a` | Single target | 5.51 | 10.62 | 5.71 | 10.81 |
| `(B.4)-b` | Simple sampling + Single target | 5.21 | 10.00 | 5.46 | 10.34 |

sampling modules. However, MACs decrease further to values of 353G and 331G, contingent on layer distribution. In essence, incorporating more lower resolution components into MR-HuBERT provides the benefit of faster inference.

Table 9 presents the ASR results for the configurations with three resolutions. Despite showing marked improvement over baselines (i.e., `HuBERT-base` and `HuBERT-base`[+]), the performance of MR-HuBERT with three resolutions isn't as robust as that of **`mono-base`**. This suggests that information from lower resolutions might not always enhance the ASR task. Given the efficiency gains observed, the inclusion of lower resolutions could be perceived as balancing efficiency against performance efficacy. It's worth noting that the performance dip observed in the three-resolution MR-HuBERT appears inconsistent with findings in (Shi et al., 2023d). The latter study revealed that features fused from multi-resolution HuBERTs across varying resolutions can bolster ASR tasks. Our hypothesis is that this performance discrepancy might stem from each resolution's constrained model capacity. A deeper dive into this is required to determine if lower resolutions can indeed boost performance.

## B.3 Simpler Upsampling & Downsampling Modules

As detailed in Section 3.3, our proposed architecture's sampling module employs a blend of upsampling and downsampling to achieve a flexible ratio between any two resolutions. However, when dealing with low resolutions that are evenly divisible by their corresponding high resolutions, there's no need to simultaneously deploy both the upsample and downsample modules. This simultaneous use introduces an unnecessary computational overhead. Given this, we delve into a more streamlined setting in this section: the upsampling module is dedicated solely to upsampling, and the downsampling module focuses only on downsampling. While this streamlined approach slightly curtails the computational load (reducing MACs from 394G to 390G) and marginally shrinks the

Table 11: Ablation study configurations focusing on singular resolution and svelte model dimensions in the *base* setting.

| Model | Layers | Resolutions (ms) | Num. Param (M) | MACs (G) |
|---|---|---|---|---|
| HuBERT-base | 12 | 20 | 95 | 431 |
| HuBERT-base$^+$ | 12 | 20 | 95 | 431 |
| **mono-base** | (4, 4, 4) | (20, 40) | 97 | 394 |
| (B.5)-a | (4, 4, 4) | (20, 20) | 97 | 439 |
| (B.6)-a | (3, 3, 3) | (20, 40) | 76 | 339 |
| (B.6)-b | (3, 3, 3) | (20, 20) | 76 | 373 |

parameter size (from 97M to 96M), it lacks the flexibility to handle unconventional ratios, such as 3:4, between resolutions.

The derived model, dubbed (B.3)-a, is subsequently fine-tuned for the ASR task, with outcomes presented in Table 10. From the results, it is evident that the MR-HuBERT equipped with the simplified sampling modules outperforms in low-resource situations, specifically the 1-hour and 10-hour ASR training scenarios. However, its performance isn't as consistent in the more extensive 100-hour experiment, particularly when juxtaposed against **mono-base**.

## B.4 SINGLE PREDICTION TARGET

As delineated in Section 3.4, our model incorporates a summation of masked unit prediction losses derived from all resolutions. In this subsection, we pivot to gauge the efficacy of deploying a singular masked unit prediction, sidelining the amalgamation of intermediate losses. Originating from **mono-base**, the resultant model, designated as (B.4)-a, benefits from an approximate reduction of 1M in parameter size. This reduction is achieved by discarding prediction heads assigned for the supplemental low-resolution masked unit prediction loss. Concurrently, we assess (B.4)-b, which melds the single prediction feature with the streamlined sampling module, as expounded upon in Appendix B.3.

Both models, (B.4)-a and (B.4)-b, have their performance metrics tabulated in Table 10. Overall, a distinct performance hierarchy emerges: (B.3)-a outstrips (B.4)-b, which in turn surpasses (B.4)-a. This sequence underscores the indispensability of the multi-task objective spanning multiple resolutions for MR-HuBERT. Moreover, when navigating models fixated on a solitary prediction target, the elementary sampling modules exhibit more potency compared to their flexible counterparts.

## B.5 SINGLE RESOLUTION

A salient feature of MR-HuBERT is its concurrent utilization of diverse resolutions. In this subsection, we distill this multifaceted design down to a singular resolution. The intention behind this simplification is to probe the contributory essence of the multi-resolution concept to the model's efficacy. We harness the architectural blueprint delineated in Section 3.2, albeit employing a consistent resolution across intermediate components. Consequently, this model forsakes the computational advantages derived from sequence reduction in self-attention calculations, culminating in a heightened computational overhead as reflected in the MACs of 439G. Intriguingly, this computational cost surpasses that of the native HuBERT, clocking in at 431G, as evidenced in Table 11.

The experimental results are cataloged in Table 12. Across the 100-hour ASR dataset, the proposed **mono-base** unambiguously outperforms its singular resolution counterpart, (B.5)-a. However, when venturing into the 1-hour and 10-hour ASR realms, the outcomes are more equivocal. Bearing both efficiency and performance in mind, these findings underscore the pivotal influence of multi-resolution strategies in bolstering MR-HuBERT's impressive performance benchmarks. Please also refer to Appendix D, where we identify more benefits from introducing multiple resolutions.

Table 12: Ablation study for singular resolution and svelte models within the *base* context. The experiments are conducted on ASR fine-tuning experiments over LibriSpeech subsets.

| Model | MACs | dev-clean | dev-other | test-clean | test-other |
|---|---|---|---|---|---|
| *1-hour labeled* | | | | | |
| HuBERT-base | 431 | 20.17 | 28.11 | 20.64 | 28.87 |
| HuBERT-base⁺ | 431 | 19.64 | 25.08 | 20.15 | 25.63 |
| **mono-base** | 394 | **18.78** | 23.72 | **19.26** | 24.46 |
| (B.5)-a | 439 | 18.87 | **23.37** | 19.69 | **24.05** |
| (B.6)-a | 339 | 18.73 | 24.40 | 19.37 | 24.78 |
| (B.6)-b | 373 | 19.41 | 25.32 | 19.67 | 26.00 |
| *10-hour labeled* | | | | | |
| HuBERT-base | 431 | 9.62 | 16.60 | 9.71 | 17.00 |
| HuBERT-base⁺ | 431 | 9.51 | 14.27 | 9.72 | 14.89 |
| **mono-base** | 394 | **8.51** | 13.18 | **8.46** | 13.51 |
| (B.5)-a | 439 | 8.56 | **12.73** | 8.69 | **12.89** |
| (B.6)-a | 339 | 9.13 | 14.87 | 9.36 | 15.22 |
| (B.6)-b | 373 | 9.13 | 14.43 | 9.38 | 14.92 |
| *100-hour labeled* | | | | | |
| HuBERT-base | 431 | 5.76 | 12.90 | 5.81 | 12.76 |
| HuBERT-base⁺ | 431 | 5.71 | 10.66 | 5.97 | 10.87 |
| **mono-base** | 394 | **4.89** | **9.04** | **4.92** | **9.17** |
| (B.5)-a | 439 | **4.89** | 9.46 | 4.93 | 9.59 |
| (B.6)-a | 339 | 5.31 | 11.07 | 5.55 | 11.19 |
| (B.6)-b | 373 | 5.31 | 11.11 | 5.47 | 11.20 |

## B.6 COMPACT MODEL

Motivated by the conspicuous performance advantage of MR-HuBERT over traditional HuBERT, we pivot our efforts towards crafting a more svelte version of MR-HuBERT, prioritizing computational economy. Eschewing the convention of a four-layer encoder, our pared-down MR-HuBERT, christened (B.6)-a, adopts a three-layer encoder scheme. This strategic recalibration augments inferential speed without significantly compromising on performance standards. The architectural nuances are delineated in Table 11. It's worth noting that our investigative purview extends to another optimized model, (B.6)-b, which amalgamates the principles of the single-resolution approach detailed in Section B.5.

As revealed in Table 12, the compact iteration understandably possesses diminished modeling prowess, translating to a performance dip relative to **mono-base**. Yet, even with this inherent constraint, it remains competitive with the original HuBERT — a noteworthy feat considering the model operates with 20% fewer parameters and realizes a 21% enhancement in inference speed.

## B.7 TARGET UNITS FOR PREDICTION

As delineated in Section 4.1, our approach favored skip-downsampling the designated high-resolution units to obtain target low-resolution units for the intermediate masked prediction supervision. This strategy emerged as the most efficacious in training MR-HuBERT effectively. Nevertheless, we ventured into exploratory ablations using alternative units. Given that direct skip-downsampling isn't inherently data-driven, we experimented with units extracted from the pre-trained 40ms-resolution HuBERT model, HuBERT-base-40, in alignment with the model architecture introduced by Shi et al. (2023d). Additionally, we leveraged units from the increasingly prevalent Encodec approach as elucidated by (Défossez et al., 2022). It's worth noting that our preliminary observations revealed suboptimal performance for most models, leading us to restrict our analysis to just the 10-hour training scenarios. Nonetheless, we present these findings to offer a repository of insights for curious researchers.

Table 13: Ablation study on different target units within the *base* context. The experiments are conducted on ASR fine-tuning experiments over LibriSpeech subsets. `HuBERT-base-40` represents a model trained on 40ms resolution, whereas `HuBERT-base`$^0$ denotes the model's first iteration trained with MFCC clusters. KM symbolizes the $K$-means algorithm with $K = 1000$, and Encodec units are denoted as Encodec-{Frequency}-{No. Stream}.

| Model | High-resolution | Low-resolution | dev-clean | test-clean |
|---|---|---|---|---|
| `HuBERT-base` | KM(`HuBERT-base`$^0$) | - | 9.62 | 9.71 |
| `HuBERT-base`$^+$ | KM(`HuBERT-base`) | - | 9.51 | 9.72 |
| **`mono-base`** | KM(`HuBERT-base`) | Skip(KM(`HuBERT-base`)) | **8.51** | **8.46** |
| `(B.7)-a` | KM(`HuBERT-base`) | KM(`HuBERT-base-40`) | 9.20 | 9.36 |
| `(B.7)-b` | Encodec-50-1 | Skip(Encodec-50-1) | 26.98 | 27.34 |
| `(B.7)-c` | Encodec-50-1 | Encodec-25-1 | 18.74 | 19.15 |
| `(B.7)-d` | Encodec-50-2 | Skip(Encodec-50-1) | 27.56 | 28.19 |

Table 14: Ablation study configurations in *large* settings. Frames/Step is shown in the format of Maximum Number of Frames * Gradient Accumulation. The Label column represents the model to extract hidden states for unit discovery. Audio Norm. is whether to conduct audio normalization to the raw audio.

| Model | Frames/Step | Label | Audio Norm. | Layers | Note | Num. Param (M) | MACs (G) |
|---|---|---|---|---|---|---|---|
| `HuBERT-large` | 90k * 1 | `HuBERT-base` | True | 24 | - | 316 | 1116 |
| `HuBERT-large`$^*$ | 90k * 1 | `HuBERT-base` | True | 24 | - | 317 | 1116 |
| **`mono-large`** | 30k * 3 | `HuBERT-base` | True | (8, 8, 8) | - | 321 | 971 |
| `(B.8)-a` | 60k * 1 | `HuBERT-base` | False | (8, 8, 8) | - | 321 | 971 |
| `(B.8)-b` | 60k * 1 | `HuBERT-base` | True | (8, 8, 8) | - | 321 | 971 |
| `(B.8)-c` | 60k * 1 | `HuBERT-large` | True | (8, 8, 8) | - | 321 | 971 |
| `(B.8)-d` | 30k * 8 | `HuBERT-large` | True | (8, 8, 8) | - | 321 | 971 |
| `(B.8)-e` | 90k * 1 | `HuBERT-base` | True | (8, 8, 8) | - | 321 | 971 |
| `(B.8)-f` | 90k * 1 | `HuBERT-large` | True | (8, 8, 8) | - | 321 | 971 |
| `(B.8)-g` | 90k * 1 | `HuBERT-base` | True | (10, 4, 10) | - | 321 | 1049 |
| `(B.8)-h` | 90k * 1 | `HuBERT-large` | True | (10, 4, 10) | - | 321 | 1049 |
| `(B.8)-i` | 80k * 1 | `HuBERT-base` | True | (8, 8, 8) | Simple Sampling | 319 | 965 |
| `(B.8)-j` | 80k * 1 | `HuBERT-large` | True | (8, 8, 8) | Simple Sampling | 319 | 965 |

Refer to Table 13 for detailed results. Interestingly, harnessing units from `HuBERT-base-40` didn't elevate performance. This leads us to conjecture that MR-HuBERT may exhibit sensitivity to the homogeneity of prediction targets spanning diverse resolutions. In the case of Encodec, the outcomes were less than stellar, suggesting that a localized acoustic discrete representation might not be synergistic with the semantic learning intricacies inherent in masked unit prediction.

## B.8 LARGE SETTINGS

In the context of *large* settings, MR-HuBERT continues to be examined. Table 14 delineates ten candidate configurations in the *large* settings. Consistently, all models are trained for 400k steps, analogous to **`mono-base`** and **`mono-large`**. These configurations not only probe further into the ablation conditions established in the *base* settings but also explore factors specifically impacting the performance of MR-HuBERT in the *large* settings. These encompass audio normalization to the raw audio, variations in batch size, and the adoption of different target unit sequences either from `HuBERT-base` or `HuBERT-large`[12]. Owing to memory constraints on V100-32GB, four models, specifically `(B.8)-e`-`(B.8)-h`, are trained on 128 A100-80GB GPUs.

The results for the ASR experiments in *large* settings are encapsulated in Table 15. A distilled account of key findings is as follows:

---

[12]Layer 9 and Layer 15 are respectively chosen for `HuBERT-base` and `HuBERT-large` for unit discovery. Post this, units are derived from the $K$-means method, with $K = 1000$.

Table 15: Ablation study in *large* settings. The experiments are conducted on ASR fine-tuning experiments over LibriSpeech subsets.

| Model | dev-clean | dev-other | test-clean | test-other |
|---|---|---|---|---|
| *1-hour labeled* | | | | |
| `HuBERT-large` | 14.42 | 18.80 | 14.40 | 19.29 |
| `HuBERT-large`[*] | 15.09 | 18.20 | 14.90 | 18.05 |
| **`mono-large`** | 6.44 | 10.94 | 6.37 | 11.41 |
| `(B.8)-a` | 20.62 | 23.43 | 20.66 | 23.45 |
| `(B.8)-b` | 7.31 | 12.58 | 7.32 | 13.39 |
| `(B.8)-c` | 7.15 | 12.30 | 7.37 | 12.89 |
| `(B.8)-d` | 6.53 | 11.79 | 6.64 | 12.14 |
| `(B.8)-e` | 6.40 | 10.89 | 6.25 | 11.03 |
| `(B.8)-f` | 6.83 | 12.26 | 6.97 | 12.77 |
| `(B.8)-g` | **6.21** | **10.21** | **6.11** | **10.63** |
| `(B.8)-h` | 6.83 | 12.52 | 6.81 | 12.63 |
| `(B.8)-i` | 6.42 | 11.29 | 6.50 | 11.91 |
| `(B.8)-j` | 6.78 | 12.06 | 6.92 | 12.53 |
| *10-hour labeled* | | | | |
| `HuBERT-large` | 5.68 | 8.67 | 5.75 | 8.96 |
| `HuBERT-large`[*] | 5.61 | 8.68 | 5.57 | 9.02 |
| **`mono-large`** | 5.58 | 8.57 | 5.52 | 8.74 |
| `(B.8)-a` | 6.07 | 8.97 | 5.89 | 9.37 |
| `(B.8)-b` | 5.93 | 8.80 | 5.87 | 9.26 |
| `(B.8)-c` | 5.79 | 8.83 | 5.79 | 9.03 |
| `(B.8)-d` | **5.48** | 8.34 | 5.48 | 8.66 |
| `(B.8)-e` | 5.73 | 8.62 | 5.62 | 8.91 |
| `(B.8)-f` | 5.68 | 8.64 | 5.52 | 8.77 |
| `(B.8)-g` | 5.58 | **8.17** | **5.41** | 8.66 |
| `(B.8)-h` | 5.49 | 8.28 | 5.45 | **8.60** |
| `(B.8)-i` | 5.77 | 8.75 | 5.63 | 8.99 |
| `(B.8)-j` | 5.66 | 8.59 | 5.64 | 9.14 |
| *100-hour labeled* | | | | |
| `HuBERT-large` | 3.11 | 6.01 | 3.14 | 6.15 |
| `HuBERT-large`[*] | 3.03 | 6.30 | 3.12 | 6.14 |
| **`mono-large`** | 3.06 | 6.04 | 3.01 | 5.98 |
| `(B.8)-a` | 3.18 | 6.31 | 3.17 | 6.30 |
| `(B.8)-b` | 3.09 | 6.01 | 3.13 | 6.13 |
| `(B.8)-c` | 3.13 | 6.11 | 3.18 | 6.17 |
| `(B.8)-d` | **2.83** | 5.86 | 2.98 | 5.91 |
| `(B.8)-e` | 3.05 | 6.27 | 3.15 | 6.02 |
| `(B.8)-f` | 2.90 | 5.90 | 3.01 | **5.74** |
| `(B.8)-g` | 2.90 | **5.64** | **2.93** | 5.88 |
| `(B.8)-h` | 2.89 | 5.71 | 3.01 | 5.69 |
| `(B.8)-i` | 3.09 | 6.22 | 3.16 | 6.13 |
| `(B.8)-j` | 2.98 | 5.94 | 3.09 | 6.02 |

- **Best performing system**: A mix of results can be discerned across LibriSpeech's four evaluation sets. However, on average, the model `(B.8)-g` stands out, chiefly due to its layer distribution modification: transitioning from the default (8, 8, 8) to (10, 4, 10). This resonates with findings in Appendix B.1, suggesting that depth isn't imperative for low-resolution modeling. Nonetheless, curtailing low-resolution layers inadvertently affects inference efficiency, as evidenced by the elevated MACs in Table 14.

- **Units from large models**: Predominantly, models trained on units from `HuBERT-large` outperform those reliant on `HuBERT-base` units. This aligns with the intuitive premise that `HuBERT-large` labels could potentially enrich the MR-HuBERT learning iteration.

Table 16: Real-time measurements on Librispeech dev-clean set.

| Model | MACs ($\downarrow$) | token_per_second ($\uparrow$) |
|---|---|---|
| `HuBERT-base` | 431 | 5833 |
| `HuBERT-large` | 1116 | 2220 |
| **`mono-base`** | 394 | 6310 |
| **`mono-large`** | 971 | 2505 |
| `(B.1)-a` | 394 | 6293 |
| `(B.1)-b` | 416 | 5911 |
| `(B.1)-c` | 394 | 6299 |
| `(B.2)-a` | 353 | 6925 |
| `(B.2)-b` | 331 | 7332 |
| `(B.2)-c` | **316** | **7580** |
| `(B.3)-a` | 390 | 6435 |
| `(B.4)-a` | 394 | 6322 |
| `(B.4)-b` | 390 | 6450 |
| `(B.5)-a` | 439 | 5229 |
| `(B.6)-a` | 339 | 7096 |
| `(B.6)-b` | 373 | 6670 |

- **Batch size matters**: Corroborating the assertions of Hsu et al. (2021a), large batch sizes appear favorable for HuBERT training. A juxtaposition of `(B.8)-b` to `(B.8)-f` indicates that augmenting the batch size can potentially bolster MR-HUBERT's performance.

- **Do use audio normalization**: Historically, audio normalization is typically applied in *large* settings of speech self-supervised learning, while it's omitted in the *base* settings. Our `(B.8)-a` model substantiates that audio normalization is quintessential for the successful training of *large* setting models on vast unlabeled datasets.

- **Simplified sampling is not recommended**: As elaborated in Appendix B.3, models employing simplified sampling modules demonstrate performance metrics closely mirroring those integrating our flexible sampling modules. However, in *large* settings, this parallelism breaks, revealing consistent enhancements when utilizing our tailored flexible sampling modules over the simplified versions.

## C    INFERENCE SPEED

Although MACs offer a theoretical estimate of execution time, they are not always a reliable indicator of actual inference speed, particularly given the parallel processing capabilities of GPUs. To address this, we conduct empirical tests to compare theoretical predictions with real-world performance. We measure the inference speed in terms of 'tokens_per_second' using Fairseq on the Librispeech dev-clean set. This measurement is the average of ten times to account for variability in real-time execution.

Our findings, detailed in Table 16, reveal that MR-HuBERT models demonstrate a significant and consistent increase in speed compared to HuBERT models in both *base* and *large* settings. Notably, the model `(B.2)-c`, equipped with three resolutions, emerges as the fastest in terms of inference speed. This empirical evidence suggests a strong alignment between the MACs calculations presented earlier and the actual performance observed in real-world scenarios.

## D    MORE IN SUPERB BENCHMARK

### D.1    SUPERB SCORE IN SUPERB BENCHMARK

The SUPERB score (i.e., SUPERB$_s$ is a sophisticated metric designed to provide a standardized assessment across various tasks, each potentially with its own scoring system (Feng et al., 2023).

Table 17: Information to calculate SUPERB score in Section 4.3. All the results are from the SUPERB leaderboard on August 15, 2023.

| Model | | | | Understanding | | | | Enhancement | | |
|---|---|---|---|---|---|---|---|---|---|---|
| | PR($\downarrow$) | ASR($\downarrow$) | IC($\uparrow$) | KS($\uparrow$) | SF-F1($\uparrow$) | SF-CER($\downarrow$) | ST($\uparrow$) | SE-STOI($\uparrow$) | SE-PESQ($\uparrow$) | SS($\uparrow$) |
| FBank | 82.00 | 23.18 | 10.44 | 8.63 | 69.64 | 52.92 | 2.32 | 0.94 | 2.55 | 9.23 |
| SOTA | 3.09 | 3.36 | 99.34 | 97.89 | 92.25 | 17.61 | 25.52 | 0.95 | 3.06 | 11.19 |

By employing linear interpolation between Mel filter banks feature (FBank) scores and state-of-the-art (SOTA) representation scores, it normalizes scores across different scales. If a single task has multiple metrics, an intra-task average is computed, ensuring that tasks with a myriad of metrics don't dominate the overall score. Subsequently, an inter-task average is derived, guaranteeing each task's equal contribution to the final score. A scaling factor of 1000 amplifies readability. For consistency, the score in this paper benchmarks against a static snapshot of the SUPERB leaderboard from August 15, 2023, as detailed in Table 17. Thoughtfully, SUPERB score's design considers task difficulty, granting more weight to tasks where even small advancements signify significant progress. This approach ensures a balanced evaluation across varying tasks, highlighting the metric's comprehensive and fair nature.

Let $\psi_{\tau,i}$ be the $i$th metrics for task $\tau$, $\psi_{\tau,i}(f)$ be the corresponding score of upstream model $f$, $\mathcal{T}$ be the set of tasks, and $I_\tau$ be the set of metrics for task $\tau$. Then, the detailed formulation is as:

$$\text{SUPERB}_s(f) = \frac{1000}{|\mathcal{T}|}\Sigma_\tau^{\mathcal{T}}\frac{1}{|I_\tau|}\Sigma_i^{I_\tau}\frac{\psi_{\tau,i}(f) - \psi_{\tau,i}(\text{FBank})}{\psi_{\tau,i}(\text{SOTA}) - \psi_{\tau,i}(\text{FBank})}. \tag{7}$$

### D.2 VOICE CONVERSION IN SUPERB BENCHMARK

In voice conversion, self-supervised learning representations have become increasingly popular as intermediate features for speech generation, as demonstrated by notable works such as (Wang et al., 2022; Huang et al., 2022b;a; 2021; Wu et al., 2022; Choi et al., 2021; Huang et al., 2023). Drawing inspiration from Tsai et al. (2022), we also extended our research to voice conversion tasks to examine the efficacy of our approach.

To achieve this, we largely followed the blueprint provided by the S3PRL recipe on the Voice Conversion Challenge 2020 (VCC2020) as detailed by (Yi et al., 2020). In particular, our experiments employed the `Taco2-AR` model as the primary downstream mechanism, a model introduced by (Liu et al., 2020b). The final waveform synthesis was facilitated by a pre-trained parallel WaveGAN-based vocoder, a method pioneered by (Yamamoto et al., 2020).

For our evaluation metrics, we leaned on Mean Cepstrum Distortion (MCD), WER for ASR, and ACC for SV, utilizing pre-trained models available within the S3PRL toolkit. Echoing the methodology behind the SUPERB score articulated in Appendix D.1, we derived a comprehensive score by averaging across all evaluation metrics.

The outcomes of these experiments are presented in Table 18. As an important side note, rather than directly referencing numbers from Tsai et al. (2022), we opted to rerun the experiments for `HuBERT-base` and `HuBERT-large`. This decision stemmed from challenges faced in replicating the original outcomes, potentially due to variations in ASR checkpoints or tweaks in hyperparameter settings. According to the results, we observe marginal improvements in the *base* setting, but worse performance in the *large* setting. Our hypothesis is that the data might suffer from overfitting issues with the enhanced modeling power of the large model. We plan to delve deeper into this in subsequent research, with the aim to better harness the capabilities of MR-HuBERT for voice conversion.

### D.3 ABLATION MODELS IN SUPERB BENCHMARK

In our aforementioned ablation studies, the evaluation was limited to the ASR performance of each model. This scope might not offer a comprehensive assessment, especially when considering the diverse objectives of different tasks. Hence, we extended our evaluation to encompass most models

Table 18: Voice conversion evaluation for the proposed method.

| Model | MCD($\downarrow$) | ASR-WER($\downarrow$) | SV-ACC($\uparrow$) | SUPERB$_{vc}$ |
|---|---|---|---|---|
| FBank | 8.47 | 38.30 | 77.25 | 0.0 |
| SOTA | 7.08 | 8.00 | 100.00 | 1000.0 |
| HuBERT-base | 7.47 | 10.93 | 97.50 | 854.6 |
| HuBERT-base[+] | 7.32 | **10.60** | 99.00 | 903.4 |
| HuBERT-large | 7.23 | 10.98 | **99.25** | 915.7 |
| HuBERT-large[*] | 7.24 | 11.53 | **99.25** | **934.6** |
| **mono-base** | **7.18** | 11.15 | **99.25** | 921.3 |
| **mono-large** | 7.56 | 11.93 | 98.50 | 851.3 |

in the SUPERB benchmark, as detailed in Appendix B. The exhaustive results are cataloged in Table 19. Below, we provide concise discussions for each task:

- **PR, KS, SF, ST, and SS**: Across these five tasks, which target understanding and enhancement, respectively, MR-HuBERT consistently outshines HuBERT. There's a noticeable performance uplift across both *base* and *large* settings, corroborated by nearly all configurations in Appendix B.

- **ASR**: In *base* settings, models tend to surpass the baselines for ASR. However, the performance landscape shifts in the *large* settings, often not in favor. Multiple factors could be responsible — perhaps the challenges of applying CTC to low-resolution, repeated features, or constraints from frozen representations. Given these observations as well as the exploration in Appendix B, a more sophisticated fusion strategy might be beneficial when leveraging MR-HuBERT as an upstream, or fine-tuning could be explored for speech recognition tasks.

- **IC**: The *base* models benefit from low-resolution data, yielding better intent classification accuracy. In contrast, despite one *large* model setting a benchmark for accuracy, many configurations don't yield improvements. A plausible cause, discerned from training curves, could be overfitting on a limited dataset. A comprehensive study on larger intent classification datasets, such as SLURP (Bastianelli et al., 2020), might offer clearer insights.

- **SE**: In *base* settings, MR-HuBERT consistently registers worse PESQ for SE, while the trend inverts in *large* settings. We theorize that MR-HuBERT initially emphasizes semantic information. But as model size increases, its augmented high-resolution encoders facilitate finer local information processing. When these high-resolution encoders robustly learn local patterns, the model's generalization capabilities arguably supersede single-resolution counterparts, like the baseline HuBERT. This conjecture is supported by the SS task, where the *large* MR-HuBERT demonstrates a significant edge over baselines, in contrast to the *base* setting.

While the preceding discussion predominantly centers on individual tasks, we consolidate categorical SUPERB scores in Table 20. In aggregate terms, the apex model—contrary to the ASR fine-tuning experiments delineated in Appendix B—is (B.8)-d, which leverages labels from HuBERT-large and employs the maximum batch size of (30k * 8 * 128) frames (amounting to approximately 1920 seconds or 0.53 hours) per step.

## D.4 LAYER WEIGHTS ANALYSIS OF SUPERB BENCHMARK

As discussed in Appendix D.3, we postulate that MR-HuBERT has implicitly prioritized different types of information across its resolutions. Intriguingly, the weighted summation approach in the SUPERB benchmark offers an insightful perspective into the layer-wise significance of the model for diverse downstream tasks. Prior works have employed these weights to ascertain the contribution of individual layers to specific downstream tasks (Chang et al., 2021; Chen et al., 2022b; Hung et al., 2022; Chen et al., 2022c; Shi et al., 2023a; Lin et al., 2023; Shi et al., 2023d; Otake et al., 2023; Chen et al., 2022a). Given that the weights of each layer participate in the backpropagation process,

Table 19: Ablation study on SUPERB Benchmark

| Model | PR(↓) | ASR(↓) | IC(↑) | KS(↑) | SF-F1(↑) | SF-CER(↓) | ST(↑) | SE-STOI(↑) | SE-PESQ(↑) | SS(↑) |
|---|---|---|---|---|---|---|---|---|---|---|
| | | | | | Understanding | | | Enhancement | | |
| | | | | | | | | | | |
| *Baseline* (Section 4.1) | | | | | | | | | | |
| `HuBERT-base` | 5.40 | 6.42 | 98.34 | 96.30 | 88.53 | 25.20 | 15.53 | **0.94** | 2.58 | 9.36 |
| `HuBERT-base`[+] | 4.56 | 6.34 | 98.39 | 96.46 | 89.12 | 23.10 | 16.33 | 0.93 | 2.55 | 9.72 |
| `HuBERT-large` | 3.54 | 3.62 | 98.76 | 95.29 | 89.81 | 21.76 | 20.01 | **0.94** | 2.64 | 10.45 |
| `HuBERT-large`[*] | 3.59 | **3.53** | 98.73 | 97.70 | 89.88 | 22.51 | 20.02 | **0.94** | 2.65 | 10.61 |
| *Proposed Method* (Section 4.1) | | | | | | | | | | |
| **`mono-base`** | 4.16 | 5.76 | 98.68 | 96.49 | 88.96 | 23.59 | 16.94 | **0.94** | 2.55 | 9.92 |
| **`mono-large`** | **3.15** | 3.78 | 98.76 | **97.76** | 90.57 | **20.60** | 21.05 | **0.94** | **2.67** | 10.97 |
| *Layer Size* (Appendix B.1) | | | | | | | | | | |
| (B.1)-a | 4.20 | 5.87 | 98.71 | 96.59 | 89.42 | 22.95 | 16.96 | **0.94** | 2.51 | 9.64 |
| (B.1)-b | 4.05 | 5.67 | 98.76 | 96.20 | 89.29 | 22.52 | 16.85 | **0.94** | 2.54 | 9.63 |
| (B.1)-c | 4.36 | 5.98 | 98.89 | 96.66 | 89.24 | 24.15 | 16.65 | **0.94** | 2.52 | 9.78 |
| *Multi-Resolution Analysis* (Appendix B.2) | | | | | | | | | | |
| (B.2)-a | 4.36 | 6.02 | 98.84 | 96.11 | 88.63 | 24.09 | 17.02 | **0.94** | 2.52 | 9.65 |
| (B.2)-b | 4.30 | 6.28 | 98.20 | 96.17 | 89.06 | 23.99 | 16.42 | **0.94** | 2.53 | 9.73 |
| (B.2)-c | 4.60 | 6.87 | 98.81 | 95.85 | 87.15 | 27.61 | 16.35 | **0.94** | 2.55 | 9.78 |
| *Simpler Upsampling & Downsampling Modules* (Appendix B.3) | | | | | | | | | | |
| (B.3)-a | 4.20 | 5.68 | 98.68 | 96.07 | 88.87 | 24.18 | 16.93 | **0.94** | 2.54 | 9.77 |
| *Single Prediction Target* (Appendix B.4) | | | | | | | | | | |
| (B.4)-a | 4.40 | 6.31 | 98.18 | 96.62 | 89.48 | 23.74 | 16.45 | **0.94** | 2.56 | 9.94 |
| (B.4)-b | 4.50 | 6.30 | 98.84 | 96.59 | 89.08 | 23.78 | 16.53 | **0.94** | 2.55 | 9.78 |
| *Single Resolution* (Appendix B.5) | | | | | | | | | | |
| (B.5)-a | 4.20 | 5.78 | 98.34 | 96.36 | 88.23 | 24.53 | 16.57 | **0.94** | 2.54 | 9.76 |
| *Compact Model* (Appendix B.6) | | | | | | | | | | |
| (B.6)-a | 4.39 | 6.25 | 98.50 | 96.17 | 88.10 | 25.41 | 15.65 | **0.94** | 2.50 | 9.80 |
| (B.6)-b | 5.09 | 6.06 | 97.94 | 95.46 | 88.67 | 24.48 | 15.51 | **0.94** | 2.53 | 9.80 |
| *Large Settings* (Appendix B.8) | | | | | | | | | | |
| (B.8)-a | 3.36 | 3.96 | 98.81 | 96.98 | 90.36 | 21.60 | 19.86 | **0.94** | 2.66 | 10.45 |
| (B.8)-b | 3.29 | 4.05 | 97.73 | 97.44 | 90.27 | 21.80 | 20.17 | **0.94** | 2.65 | 10.53 |
| (B.8)-c | 3.37 | 4.01 | 98.68 | 97.60 | 89.95 | 21.74 | 19.60 | **0.94** | 2.66 | 10.80 |
| (B.8)-d | 3.21 | 3.68 | 98.76 | 97.60 | 90.36 | 21.25 | **21.52** | **0.94** | **2.67** | **11.25** |
| (B.8)-e | 3.46 | 4.06 | 98.39 | 97.34 | 90.57 | 21.26 | 20.23 | **0.94** | 2.65 | 10.91 |
| (B.8)-f | 3.40 | 4.02 | **99.05** | 97.50 | 89.89 | 21.54 | 20.56 | **0.94** | 2.66 | 10.93 |
| (B.8)-g | 3.21 | 3.81 | 98.42 | 97.31 | 90.22 | 21.36 | 20.50 | **0.94** | **2.67** | 10.83 |
| (B.8)-h | 3.18 | 3.92 | 98.39 | 97.14 | **90.64** | 20.65 | 20.43 | **0.94** | 2.66 | 10.72 |
| (B.8)-i | 3.41 | 4.15 | 98.39 | 97.14 | 89.95 | 22.28 | 19.89 | **0.94** | 2.65 | 10.97 |
| (B.8)-j | 4.93 | 3.99 | 98.34 | 97.27 | 89.64 | 22.45 | 20.17 | **0.94** | 2.66 | 10.91 |

we surmise these weights can elucidate how each layer contributes to the final prediction in relation to the training objectives of each task.

Observing distinct behaviors between the models in both *base* and *large* configurations, we conduct separate comparisons for these two settings:

***Base* setting**: The juxtaposition of **`mono-base`** with `HuBERT-base` is illustrated in Figure 3.[13] In this comparison, both models manifest analogous behaviors. Broadly, echoing previous findings (Chen et al., 2022a; Chang et al., 2021; Chen et al., 2022b), layer weights are notably task-dependent: understanding tasks predominantly engage the later layers while enhancement tasks favor the initial layers.

Yet, distinct layer weight distributions are palpable:

- For the ASR task, while `HuBERT` predominantly targets its bottom layers (layers 9-11), **`mono-base`** allocates over 40% of its attention to low-resolution layers 8 and 9. This inclination is explicable given the rich semantic content of low-resolution layers. This trait elucidates the pronounced contribution of layers 11-12 in MR-HuBERT for the PR

---

[13]To clarify the distinction in layer numbers, MR-HuBERT encompasses not just the Transformer layers but also the outputs of the sampling module. Consequently, a two-resolution MR-HuBERT introduces two extra layers into the weighted summation computation during SUPERB downstream task training. Specifically, for ***mono-base***, the low-resolution layers span from layer 6 to layer 10.

Table 20: Ablation study in categorical SUPERB score on SUPERB Benchmark

| Model | Understanding | Enhancement | General |
|---|---|---|---|
| ***Baseline*** (Section 4.1) | | | |
| `HuBERT-base` | 861.2 | 98.20 | 670.4 |
| `HuBERT-base`[+] | 876.9 | 150.2 | 695.2 |
| `HuBERT-large` | 932.6 | 456.0 | 813.4 |
| `HuBERT-large`[*] | 936.2 | 501.5 | 827.5 |
| ***Proposed Method*** (Section 4.1) | | | |
| **`mono-base`** | 885.8 | 195.0 | 708.7 |
| **`mono-large`** | 949.7 | 609.5 | 864.6 |
| ***Layer Size*** (Appendix B.1) | | | |
| `(B.1)-a` | 888.4 | 80.4 | 686.4 |
| `(B.1)-b` | 889.5 | 127.7 | 699.1 |
| `(B.1)-c` | 881.9 | 134.8 | 695.1 |
| ***Multi-Resolution Analysis*** (Appendix B.2) | | | |
| `(B.2)-a` | 881.0 | 101.7 | 686.2 |
| `(B.2)-b` | 875.4 | 140.6 | 691.7 |
| `(B.2)-c` | 854.2 | 145.4 | 677.0 |
| ***Simpler Upsampling & Downsampling Modules*** (Appendix B.3) | | | |
| `(B.3)-a` | 883.8 | 147.0 | 699.6 |
| ***Single Prediction Target*** (Appendix B.4) | | | |
| `(B.4)-a` | 878.0 | 219.5 | 713.4 |
| `(B.4)-b` | 878.1 | 184.1 | 704.6 |
| ***Single Resolution*** (Appendix B.5) | | | |
| `(B.5)-a` | 877.1 | 163.3 | 698.7 |
| ***Compact Model*** (Appendix B.6) | | | |
| `(B.6)-a` | 863.6 | 133.0 | 680.9 |
| `(B.6)-b` | 864.6 | 156.8 | 687.7 |
| ***Large Settings*** (Appendix B.8) | | | |
| `(B.8)-a` | 934.6 | 487.4 | 822.8 |
| `(B.8)-b` | 934.3 | 484.3 | 821.8 |
| `(B.8)-c` | 931.4 | 565.2 | 839.9 |
| `(B.8)-d` | **951.1** | **686.4** | **885.0** |
| `(B.8)-e` | 937.7 | 596.8 | 852.5 |
| `(B.8)-f` | 938.9 | 584.3 | 850.3 |
| `(B.8)-g` | 940.8 | 576.9 | 849.8 |
| `(B.8)-h` | 942.2 | 543.8 | 842.6 |
| `(B.8)-i` | 929.6 | 598.5 | 846.8 |
| `(B.8)-j` | 928.3 | 598.4 | 845.8 |

task. Another distinction emerges in the SF task, where the semantic-centric slots in MR-HuBERT exhibit a propensity for low-resolution representations.

- As tasks shift their focus away from native high-resolution representations, MR-HuBERT adeptly diminishes extraneous information, hinting at a potential for implicit speech disentanglement. For tasks like SE and SS, which emphasize local acoustics, the associated downstream models discern that the information from low-resolution layers (i.e., layers 5-9) isn't advantageous and pivot their attention to earlier layers—a contrast from the more varied layer focus observed in HuBERT.

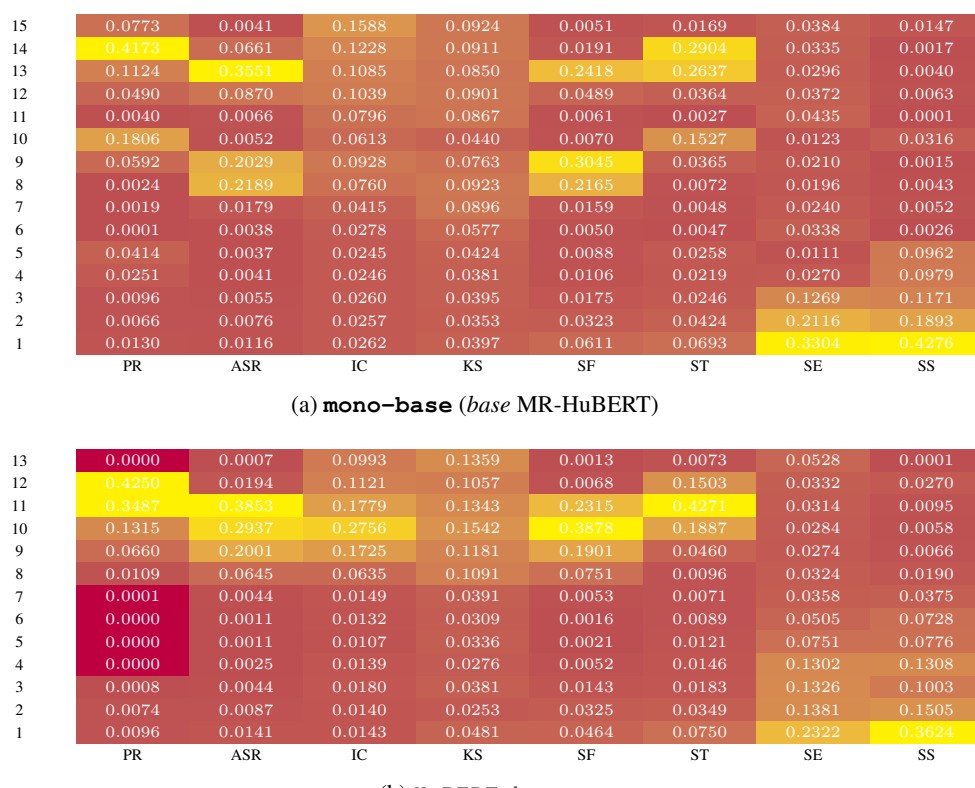

Figure 3: Layer-weight analysis on SUPERB tasks over two *base* models. The weights are the layer-wise weights after the Softmax function, which are trained together with downstream models as detailed in Section 4.3.

**Comparison in the *large* setting**: The behavior of models in the *large* setting contrasts significantly with that in the *base* setting. Figure 4 illustrates the comparison for *large* models.[14] We begin by evaluating each model individually before delving into a comparative analysis:

Three primary patterns emerge for **mono-large** when applied to SUPERB evaluation (See Figure 4a):

- **High-resolution encoder emphasis**: For tasks like SE and SS, which are associated closely with original audio signals, the first high-resolution encoder predominantly contributes.
- **Low-resolution encoder emphasis**: Understanding tasks such as PR, ASR, SF, and ST predominantly lean on the second low-resolution encoder. Nonetheless, some information from the high-resolution encoders also plays a role, particularly when predictions align sequentially and emphasize semantic content.
- **Equitable encoder distribution**: Tasks like IC and KS exhibit a balanced weight distribution across various encoders. Intriguingly, all these tasks revolve around speech classification.

For HuBERT-large, we discern three distinct trends (refer to Figure 4b):

- **Top Layer Emphasis**: Tasks such as SE are heavily reliant on the top layers.
- **Bottom Layer Emphasis**: Tasks including PR, ASR, SF, ST, and SS predominantly focus on the bottom layers.
- **Diverse Layer Influence**: Tasks like IC and KS exhibit varied focus across different layers.

---

[14]Recall from our discussion on the *base* setting, MR-HuBERT incorporates two additional layers in the final prediction, positioning the low-resolution representations between layer 10 to layer 18.

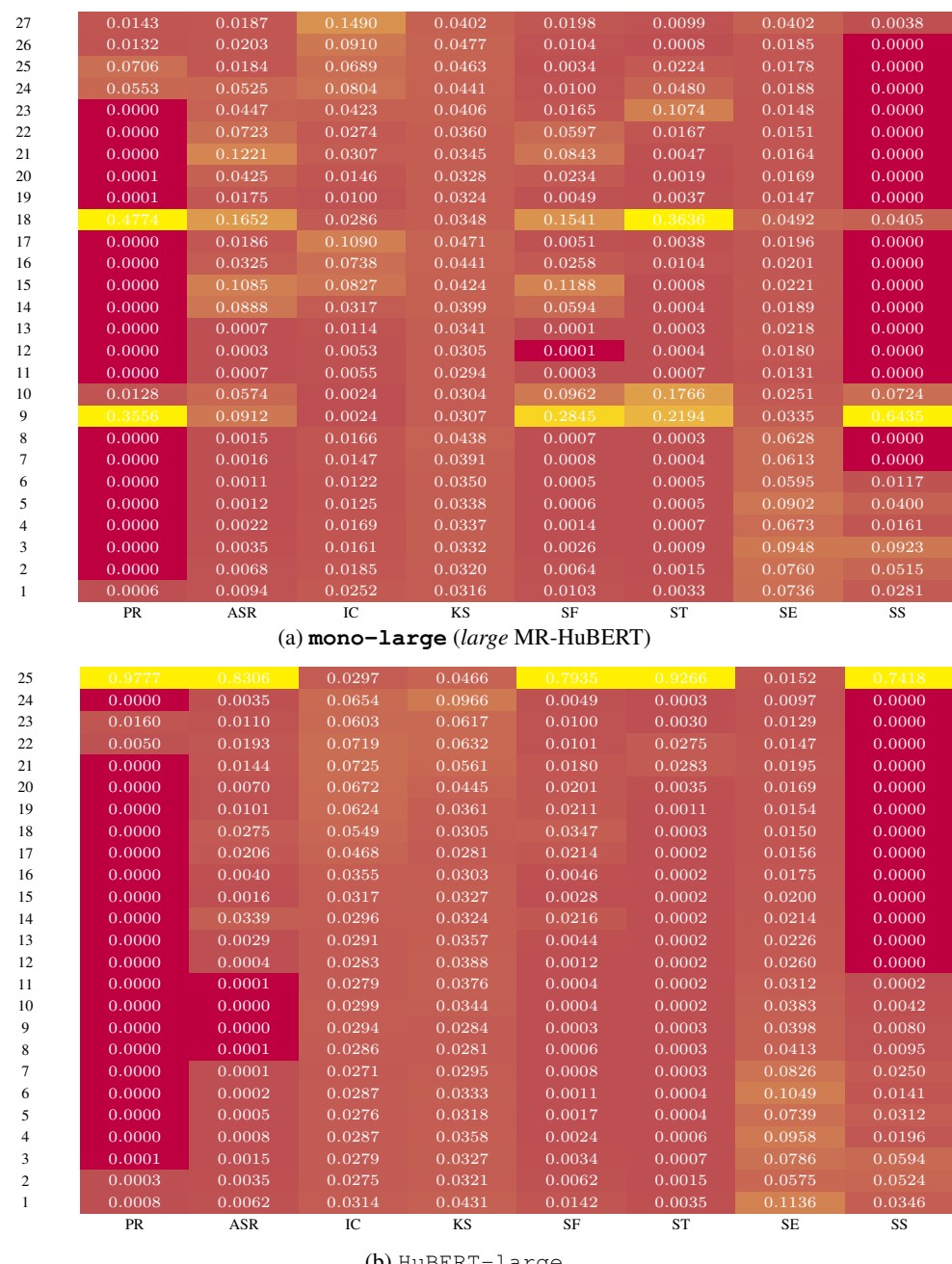

(a) **mono-large** (*large* MR-HuBERT)

(b) `HuBERT-large`

Figure 4: Layer-weight analysis on SUPERB tasks over two *large* models. The weights are the layer-wise weights after the Softmax function, which are trained together with downstream models as detailed in Section 4.3.

As for comparison, MR-HuBERT showcases a more nuanced understanding of speech signal intricacies, suggesting an implicit speech disentanglement. Conversely, HuBERT displays a rather arbitrary layer weight distribution. For instance, there's a pronounced emphasis on the final layer output for both understanding and speech separation tasks. The weight distribution patterns of MR-HuBERT hint at its potential to seamlessly transition into a more interpretable framework for speech representation studies.

Table 21: Comparison of models within the ML-SUPERB evaluation.

| Model | Num. Params (M) | Pre-Training | |
| --- | --- | --- | --- |
| | | Num. Hours | Num. Languages |
| *wav2vec2-based* | | | |
| wav2vec2-base (Baevski et al., 2020) | 95 | 1k | 1 |
| wav2vec2-large (Baevski et al., 2020) | 317 | 60k | 1 |
| robust-wav2vec2-large (Hsu et al., 2021b) | 317 | 65k | 1 |
| wav2vec2-base-23 (Wang et al., 2021a) | 95 | 100k | 23 |
| wav2vec2-large-23 (Wang et al., 2021a) | 317 | 100k | 23 |
| XLSR-53 (Conneau et al., 2020) | 317 | 56k | 53 |
| XLSR-128 (Babu et al., 2021) | 317 | 400k | 128 |
| *HuBERT-based* | | | |
| HuBERT-base (Hsu et al., 2021a) | 95 | 1k | 1 |
| HuBERT-base[+] | 95 | 1k | 1 |
| HuBERT-large (Hsu et al., 2021a) | 317 | 60k | 1 |
| HuBERT-large[*] (Hsu et al., 2021a) | 317 | 60k | 1 |
| HuBERT-base-cmn[15] | 95 | 10k | 1 |
| HuBERT-large-cmn | 317 | 10k | 1 |
| mHuBERT-base (Lee et al., 2022b) | 95 | 14k | 3 |
| mHuBERT-base[*] | 95 | 100k | 23 |
| **mono-base** | 97 | 1k | 1 |
| **mono-large** | 97 | 60k | 1 |
| **multi-base** | 97 | 100k | 23 |

# E DELVING DEEPER INTO THE ML-SUPERB BENCHMARK

## E.1 ML-SUPERB SCORE IN ML-SUPERB BENCHMARK

The ML-SUPERB score is derived as a linear-scaled average score of tasks spanning two specific leaderboards: the 10-minute and 1-hour leaderboards. Its computation is akin to that of the SUPERB score. Here, the scaling boundaries are defined by the FBank and the SOTA models. To ensure uniformity and to provide a holistic view of individual model performance, we reference the same leaderboard from the original ML-SUPERB paper when calculating the ML-SUPERB score (Shi et al., 2023a).

The ML-SUPERB benchmark encompasses a diverse spectrum of models, each pre-trained with distinct configurations (Shi et al., 2023a). To render a comprehensive view of how our method stacks up against the competition, we amalgamated our data tables with the original ML-SUPERB leaderboard. The consolidated table, Table 21, offers insights into specific model configurations, highlighting their model parameters, pre-training data size, and linguistic diversity during pre-training. Previous studies on multilingual modeling underscore the advantage of a broader language spectrum (Hou et al., 2020; Watanabe et al., 2017; Zhang et al., 2022a; Chen et al., 2023b; Toshniwal et al., 2018; Li et al., 2019a; Gaur et al., 2021; Lugosch et al., 2022; Shi et al., 2023a). Keeping this in mind, we've distinguished models based on their linguistic expanse: monolingual (blue), regional-multilingual (teal), and global-multilingual (yellow).

Table 22 provides an in-depth overview of performance metrics across benchmark tasks. To sum it up, our MR-HuBERT makes a commendable mark amidst the broader ML-SUPERB landscape. Within the monolingual category, our model conspicuously outpaces competitors—be it wav2vec2-based or HuBERT-aligned. Intriguingly, it even surpasses several multilingual counterparts, including the likes of wav2vec2-base-23, wav2vec2-large-23, and XLSR-53. This is particularly noteworthy given that these models benefit from vast datasets and broader linguistic diversity.

Navigating to the multilingual segment, our MR-HuBERT **multi-base** showcases a performance nearly on par with the frontrunner, XLSR-128, excelling in the 10-minute benchmark while slightly trailing in the 1-hour category. These outcomes are indeed remarkable, especially when accounting for our model's leaner parameters, compact pre-training data size, and reduced linguistic breadth. We anticipate MR-HuBERT to be instrumental in sculpting the future of multilingual modeling.

Table 22: Complete ML-SUPERB Benchmark Results.

| SSL | Monolingual ASR | Multilingual ASR | | LID | Multilingual ASR + LID | | | SUPERB$_s$ |
|---|---|---|---|---|---|---|---|---|
| | | Normal | Few-shot | Normal | Normal | | Few-shot | |
| | CER/PER | CER | CER | ACC | ACC | CER | CER | |
| *ML-SUPERB Benchmark* (Shi et al., 2023a) | | | | | | | | |
| FBank | 72.1 / 63.7 | 62.4 / 59.3 | 58.3 / 57.4 | 11.1 / 9.3 | 35.9 / 43.5 | 62.0 / 58.6 | 58.9 / 58.1 | 0 / 0 |
| wav2vec2-base | 44.2 / 35.9 | 43.0 / 35.5 | 45.7 / 44.3 | 54.4 / 80.8 | 66.9 / 83.6 | 40.6 / 32.1 | 44.2 / 42.6 | 755.2 / 827.2 |
| wav2vec2-large | 42.0 / 35.4 | 42.6 / 35.7 | 45.8 / 43.9 | 30.9 / 8.0 | 54.6 / 78.2 | 45.5 / 34.7 | 50.3 / 42.2 | 598.3 / 586.9 |
| robust-wav2vec2-large | 44.4 / 35.7 | 40.1 / 31.1 | 45.4 / 42.2 | 50.8 / 72.1 | 33.1 / 62.9 | 38.6 / 33.7 | 44.9 / 46.0 | 680.3 / 768.6 |
| wav2vec2-base-23 | 49.2 / 35.1 | 37.7 / 32.0 | 43.4 / 42.2 | 58.7 / 71.9 | 45.1 / 66.3 | 37.2 / 30.9 | 44.3 / 43.0 | 735.7 / 798.0 |
| wav2vec2-large-23 | 42.0 / 34.2 | 42.1 / 35.3 | 44.3 / 42.4 | 1.1 / 64.2 | 21.8 / 49.7 | 43.4 / 35.2 | 46.1 / 43.1 | 433.8 / 724.9 |
| XLSR-53 | 49.5 / 34.9 | 33.9 / 26.9 | 43.6 / 40.6 | 6.6 / 87.1 | 45.6 / 76.9 | 33.4 / 28.6 | 43.2 / 44.6 | 528.8 / 894.9 |
| XLSR-128 | 39.7 / **30.6** | **29.2 / 22.0** | 40.9 / 39.3 | **66.9 / 87.9** | 55.6 / 85.6 | **28.4 / 22.9** | 42.1 / 42.4 | 947.5 / **996.0** |
| HuBERT-base | 42.8 / 35.3 | 39.8 / 31.4 | 44.5 / 42.7 | 61.2 / 86.1 | 71.5 / 86.0 | 39.2 / 30.9 | 43.8 / 41.8 | 831.9 / 884.9 |
| HuBERT-large | **38.2** / 32.2 | 44.4 / 37.7 | 48.2 / 43.5 | 46.5 / 64.1 | 55.4 / 77.7 | 45.6 / 35.1 | 49.3 / 42.2 | 678.7 / 783.6 |
| HuBERT-base-cmn | 43.1 / 35.3 | 40.8 / 31.4 | 45.4 / 42.7 | 49.3 / 86.1 | **75.1** / 86.1 | 37.7 / 30.9 | 43.5 / 41.8 | 779.0 / 810.2 |
| HuBERT-large-cmn | 39.4 / 32.2 | 42.6 / 37.7 | 45.8 / 43.5 | 39.5 / 64.1 | 66.4 / 77.7 | 41.9 / 35.1 | 45.2 / 42.2 | 715.4 / 783.6 |
| mHuBERT-base | 41.0 / 33.0 | 40.5 / 33.4 | 45.6 / 43.6 | 52.4 / 72.5 | 46.6 / 70.9 | 36.8 / 29.7 | 44.2 / 43.1 | 746.2 / 812.7 |
| *Additional Baseline* (Section 4.1) | | | | | | | | |
| HuBERT-base[+] | 42.9 / 35.3 | 41.5 / 31.2 | 45.8 / 42.8 | 63.8 / 81.9 | 70.1 / 85.8 | 39.6 / 31.3 | 44.6 / 40.7 | 819.1 / 875.8 |
| HuBERT-large[*] | 41.2 / 32.6 | 42.8 / 32.8 | 45.6 / 42.5 | 42.3 / 58.9 | 59.2 / 84.7 | 42.3 / 29.8 | 44.1 / 41.4 | 704.5 / 817.6 |
| mHuBERT-base[*] | 40.1 / 32.3 | 36.3 / 27.3 | **38.6** / 39.0 | 64.0 / 82.0 | 70.4 / 84.6 | 35.4 / 27.1 | **39.0** / 37.0 | 950.8 / 964.5 |
| *Proposed Method* (Section 4.1) | | | | | | | | |
| **mono-base** | 42.8 / 34.6 | 40.2 / 30.6 | 45.0 / 42.2 | 67.2 / 86.3 | 68.7 / **86.9** | 40.3 / 30.6 | 44.1 / 41.6 | 843.5 / 899.9 |
| **mono-large** | 40.5 / 32.0 | 38.9 / 29.4 | 42.7 / 40.5 | 45.1 / 75.4 | 67.6 / 85.9 | 39.0 / 29.7 | 43.8 / 40.8 | 785.2 / 905.4 |
| **multi-base** | 38.3 / **30.6** | 34.1 / 27.5 | 39.6 / **38.9** | 64.0 / 85.1 | 69.9 / 84.4 | 34.4 / 28.0 | 40.9 / **36.6** | **957.2** / 986.8 |

## F  LIMITATIONS

**Dependency on Prior Models**: Instead of training from scratch, MR-HuBERT is predominantly trained using additional iterations from HuBERT discrete units. The potential of training MR-HuBERT from scratch, without leveraging previously trained models, remains unexplored.

**Performance Gaps in Specific Tasks**: While MR-HuBERT exhibits superior results compared to HuBERT, it lags behind WavLM, especially in enhancement tasks within the SUPERB framework (Chen et al., 2022a). The disparity might stem from differences in the training data and conditions. Notably, WavLM benefits from training on augmented unlabeled datasets that incorporate noise and other speech augmentations. Merging the MR-HuBERT framework with WavLM's training approach is a promising direction that warrants further investigation.

**Applicability to Non-Speech Audio Tasks**: Since MR-HuBERT's training centers around speech data, its efficacy diminishes for non-speech audio tasks, such as music or generic audio processing (Li et al., 2023b; Turian et al., 2022; Liu et al., 2022; Yuan et al., 2023; Ma et al., 2023). This limitation surfaces when trying to deploy MR-HuBERT in contexts divergent from speech. Delving into a more holistic representation is crucial to achieve peak performance in a broad spectrum of audio tasks.

## G  POTENTIAL EXTENSIONS

In our examination, MR-HuBERT emerges as a promising alternative to existing speech pre-trained models. The outcomes highlight not only its immediate relevance but also hint at a host of future research directions:

- **Integration with Other Frameworks:** While MR-HuBERT primarily hinges on the HuBERT-style training, its multi-resolution architecture can potentially be fused with a variety of self-supervised frameworks, such as wav2vec2, WavLM, w2v-bert, w2v-bert2, and data2vec (Baevski et al., 2020; Chen et al., 2022a; Chung et al., 2021; Barrault et al., 2023; Baevski et al., 2022).

- **Diverse Resolutions:** Our experimental paradigm predominantly hinged on two-resolution MR-HuBERT, albeit with a cursory glance at a three-resolution approach. Delving deeper into varying resolution combinations might unearth optimal configurations tailored to specific use cases, such as higher resolutions for detailed acoustic analysis or lower resolutions for environmental information.

- **Richer Representation:** HuBERT is renowned for its wide usage for extracting discrete semantic representations, facilitating tasks like resynthesis, voice conversion, and speech-to-speech translation (Li et al., 2023a; Huang et al., 2021; Sicherman & Adi, 2023; Polyak et al., 2021; Lee et al., 2022a; Lakhotia et al., 2021; Shi et al., 2021; Lin et al., 2022a; Lian et al., 2022; Nguyen et al., 2023; Shi et al., 2023e; Choi et al., 2023; Inaguma et al., 2023; Barrault et al., 2023; Yan et al., 2023; Huang et al., 2023). As MR-HuBERT melds low-resolution layers for enriched semantics with high-resolution layers for nuanced acoustics, it can offer a more holistic representation. This multi-faceted view could be pivotal in enhancing speech quality in generative tasks.

- **Speech Disentanglement:** Our insights, as dissected in Appendix D, highlight an implicit speech disentanglement capability in the *large* MR-HuBERT model. Scaling up the model could amplify this characteristic. Furthermore, incorporating adversarial elements can engender explicit disentanglement, proving invaluable for tasks that necessitate isolating semantic or acoustic information from speech signals. We believe the architecture would be even better integrated with existing disentanglement approaches for self-supervised learning (Qian et al., 2022; Chang et al., 2023).

