# OpenReview forum: "Multi-resolution HuBERT: Multi-resolution Speech Self-Supervised Learning with Masked Unit Prediction"
_ICLR.cc/2024/Conference — ICLR 2024 spotlight_

### Official Review · Reviewer_nPyR · 2023-10-29

**Soundness:** 4 excellent
**Presentation:** 3 good
**Contribution:** 4 excellent
**Rating:** 8
**Confidence:** 4

**Summary:**

This paper proposed to improve the self-supervised learning (SSL) method for speech signal by applying the multiple resolution processing. The motivation is to capture varying levels of information present at different resolution of speech signal. Specifically, a hierarchical Transformer is incoperated into the HuBERT-style masked prediction based model architecture. Experimental results on LibriSpeech, SUPERB and ML-SUPERB demonstrateds superior performance compared to the original HuBERT method.

**Strengths:**

1, This paper proposed new innovative method to deal with a classical problem. Self-supervised learning (SSL) has been widely studied in the past several years to leverage the unlabeled data for deep learning. This paper tackled the SSL probelm from the model architcture aspect that performed the task of multi-resolution masked units prediction. As speech signal carries both short-term and long-term characterstics, e.g., semantic level, acoustic level, etc., applying multi-resolution processing is indeed a reasonable way to analyze speech signal. While it's been applied in other domain of speech area, this paper applies it on SSL for the first time. Their contribution not only comply with this paper, but also opens a door for more potential work in this area in the future.

2, The experiments are comprehensive, clearly demonstrating the effectivness of the proposed method.

**Weaknesses:**

The writing/presentation of the paper could be improved.

**Questions:**

1, Have you or are you considering to evaluate your method on the ASR accuracy on the real conversational data, e.g., AMI, ICSI, Ali-meeting, etc. ? If you have already done with these evaluations, how is the performance compared with the original HuBERT method?
2, In the paper, it seems two types of resolution are adopted in the method. What the performance will be if you increaes the resolution types to 3 or more?

---

> ### Author Response · Authors · 2023-11-11
> **Response to Reviewer nPyR**
>
> Dear Reviewer nPyR,
>
> Thank you for your supportive feedback on our work. We are pleased to see the alignment in our perspectives on the potential future trajectories of our research. In terms of writing and presentation, we are diligently integrating the collective advice from all reviewers, including notation linkage between figure and content, rectifying formulation errors, fixing missed descriptions, and elucidating experimental setups. Should you have any particular recommendations for further improvement, we would be eager to hear them.
>
> Addressing your queries:
>
> - We have not yet evaluated our system with real conversational data as it was beyond the scope of our initial focus. Indeed, compared to WavLM, which benefits from additional and noisy data augmentation, our system might not currently perform as well. We recognize the significance of this direction and intend to explore it in subsequent research.
> - Concerning your question about multiple resolutions, we direct you to Appendix B.2. Here we assessed the impact of incorporating three resolutions. The breadth of the investigation necessary to thoroughly examine this, including the consideration of more aggressive utterance-level embeddings and finer 10ms resolutions, would be vast. Given this, we have earmarked the exploration of diverse resolutions as a future direction, as noted in Appendix F.
>
> We are grateful for your thoughtful review and hope that our responses shed light on the aspects you've inquired about. We remain committed to advancing our manuscript with the insights gained from your input.

---

### Official Review · Reviewer_hKsL · 2023-11-04

**Soundness:** 3 good
**Presentation:** 3 good
**Contribution:** 3 good
**Rating:** 8
**Confidence:** 3

**Summary:**

The authors investigate a multi-resolution evolution of HuBERT, MR-HuBERT, which augments the HuBERT architecture by integrating a lower time resolution (40 ms) transformer into the model, in addition to the standard higher resolution (20ms) processing (c.f. Figure 1).

Extensive experiments on the well known LibriSpeech, SUPERB, and ML-SUPERB datasets are conducted, and indicate that MR-HuBERT generally performs on-par or better than HuBERT, and significantly better on LibriSpeech when the amount of labelled data is very limited (1 hour).

In addition, inference speed in terms of Multiply-Add Cumulations (MACs) is reduced by 9% and 13% by the base and large variants of MR-HuBERT relative to HuBERT.

**Strengths:**

- To the best of my knowledge, MR-HuBERT is the first approach to explicitly address the integration of multi-resolution information into the pre-training of a single model, as claimed.
- Solid performance relative to HuBERT, with strong gains over appropriate baselines when limited task data is available.
- Extensive evaluations on important datasets, at multiple operating points in terms of masked pre-training (e.g. 60K vs 960 hrs) and labeled data (e.g. 1,10,100 hrs of Librispeech).
- Extensive appendix with detailed additional results and ablations.
- Code and models will be publicly released.

**Weaknesses:**

- Somewhat lower in ML novelty, as a more straightforward evolution of HuBERT.
- As acknowledged (limitations, appendix E), MR-HuBERT was not trained on augmented data like WavLM, leaving this as future work, and so performance lags behind WavLM.
- The important section on MR-HuBERT's architecture (3.2) could be improved. The processing steps are adequately described, and Figure 1 is for the most part clear enough, but with several functions f and outputs H, the relation between figure 1 and the description could be better---is equation 2 correct? Also, the operators in (eq. 2) and (eq. 3) should be introduced, as should $\phi$. Are the high resolution encoders in figure 1 the same?
- MACs are not usually a strong indication of inference speed on GPUs. This should be quantified further and/or perhaps de-emphasized in the abstract, as appropriate.
- minor: speech Self-Supervised Learning (SSL) models -> Self-Supervised Learning (SSL) speech models
- minor: we evaluate MR-HuBERT at two resolutions -> we evaluate a two resolution variant of MR-HuBERT

**Questions:**

See previous section.

---

> ### Author Response · Authors · 2023-11-11
> **Response to Reviewer hKsL**
>
> Dear Reviewer hKsL,
>
> Thank you for the encouraging remarks on our work and for your constructive feedback.
>
> In response to your points:
>
> - We concur that the core concept behind our approach is straightforward. Nevertheless, we are thrilled about the introduction of MR-HuBERT to the research community. We believe that it can unlock numerous novel pathways in the field of speech representation learning, as thoroughly discussed in Appendix F of our paper.
>
> - With regards to the performance comparison with WavLM, we acknowledge that since we have not replicated the WavLM setups precisely, including data augmentation and the use of additional datasets such as Gigaspeech&Voxpopuli, a direct comparison is not feasible at this stage. We aim to explore these enhancements in our future research.
>
> - We appreciate your attention to detail concerning the notation in our formulation. We will make the necessary corrections to our figures to improve clarity, including the reversal of $f_1^q$ and $f_2^q$. We also apologize for the oversight of not including the definition of $\phi$ in the manuscript. This will be corrected in our revised edition. Regarding the high-resolution encoders depicted in Figure 1, we will clarify that different $f_1^q$ and $f_3^q$ functions represent them, which should resolve any confusion.
>
> - Regarding the usage of MACs, your observation is astute. Our empirical findings did indicate that the actual execution time (measured by tokens_per_second in fairseq) aligns with the MACs calculations. However, due to the inherent variability in real-time measurements, we opted to present MACs as they provide a theoretical and more consistent metric. Taking your suggestion into account, we will include the real inference time from our Librispeech fine-tuning experiments in an appendix. For your immediate reference, here is a preliminary overview of the speed improvements on Librispeech dev-clean inference:
>
> | Model         | Tokens_per_second (the higher the better) |
> |---------------|-------------------|
> | hubert-base   | 5841.6            |
> | hubert-large  | 2226.2            |
> | mono-base     | 6332.3            |
> | mono-large    | 2522.1            |
>
> - Finally, we are grateful for your suggestions regarding writing improvements. These will be addressed in our forthcoming submission.
>
> Your feedback has been instrumental in enhancing the quality of our manuscript and thank you again for your effort in reviewing this paper.

---

> > ### Comment · Reviewer_hKsL · 2023-11-16
> >
> > Thank you for the response authors. These updates will resolve my concerns.

---

### Official Review · Reviewer_vPyr · 2023-11-04

**Soundness:** 4 excellent
**Presentation:** 3 good
**Contribution:** 3 good
**Rating:** 8
**Confidence:** 3

**Summary:**

The authors propose an extension of HuBERT armed with multi-resolution perception and understanding capability. The authors show the proposed methods generally outperform HuBERT with sizable improvements in, if not full stack, a wide range of speech tasks.

**Strengths:**

The idea of multi-resolution encoder makes a lot of sense as acoustic concepts typically happen with different rates, and different speech tasks also require features with diverse granularity. According to the self-contained review on related works, this work, if not the first one, is among the early explorations on using multi-resolution encoder for self-supervised speech representation learning. Some bullet points:

1. The proposed method achieved sizable improvement in the SuperB evaluation series.

2. The proposed method exhibits computational efficiencies, specifically a 9-13% reduction in computation.

3. The authors conducted extensive ablation studies to understand the effectiveness of different components. Detailed hyper parameters are also shared for reproducing purposes.

**Weaknesses:**

My major concerns are:

1. The ASR performance is not really better comparing to HuBERT.

-- According to Table one, the proposed method is better than HuBERT when the hours of labeled speech is no more than 10; However, as we scaled up the labeled speech, HuBERT shows better performance in dev.

-- According to SuperB evaluation, HuBERT large is still better (though not much) compared to the proposed method.


2. One major invention, the sampling module that employs a blend of upsampling and downsampling to achieve flexible ratio between any two resolutions, do not show clear benefits when compared to just simple sampling module.

-- According to Table 10, the proposed sampling strategy only becomes more powerful when using 100 hour of labeled speech, and the benefit is not significant. When only using 1 and 10 hour of labeled speech, the simple sampling strategy is actually doing better in terms of ASR.

-- According to Table 18, it shows different design choices and configurations would clearly affect the downstream performance, and there is on clear winner.

**Questions:**

I agree that the multi-resolution ideas is interesting, and the authors’ model do achieve very promising performance according to SuperB evaluation. My main questions have been posted in 'Weakness' section.

---

> ### Author Response · Authors · 2023-11-11
>
> Dear Reviewer vPyr,
>
> We are grateful for your positive evaluation and the attention to detail you have demonstrated in reviewing the appendix results of our paper.
>
> Regarding the issues you have raised:
>
> - Concerning ASR performance, we acknowledge that MR-HuBERT does not unequivocally surpass the original HuBERT in every test case, although it does in the majority. As we mentioned in the abstract, MR-HuBERT "exhibits superior or comparable performance to the original HuBERT."
>
>   - Specifically, with the Librispeech experiments, we have observed a marginal degradation in the dev_other set compared to the publicly available pre-trained HuBERT, yet MR-HuBERT shows better performance across the remaining sets. It is essential to recognize the inherent randomness in HuBERT training, especially from K-means clustering, which can lead to variations in performance. For a fair comparison, we have also pre-trained a \texttt{HuBERT-large}* using identical K-means clusters as MR-HuBERT. The results in Table 1 indicate that our \texttt{mono-large} model outperforms \texttt{HuBERT-large}* in all aspects except for a negligible difference in the dev_clean set.
>
>   - In the SUPERB benchmark, we have noticed the ASR performance is not as robust compared to the original HuBERT models. We have elaborated on potential reasons for this in Appendix C.3 (\textbf{ASR} bullet), and we regard this as an avenue for future improvements to MR-HuBERT. The comprehensive explanation provided in the appendix should serve to clarify this point further.
>
> - We are thankful for your scrutiny of our detailed ablation analysis. While there is some divergence in model performance across the base setting in Librispeech experiments, the flexible up/downsampling module demonstrates more consistent results in large-scale experiments and the SUPERB benchmark. We encourage you to refer to Appendix B.8 (\textbf{Simplified sampling is not recommended} bullet) and the results depicted in Table 18 for models \texttt{(B.8)-i} and \texttt{(B.8)-j}. Based on these observations, we maintain that our proposed flexible up/downsampling module is the more reliable choice over its simplified counterpart.
>
> We appreciate your thorough examination of our ablations and insightful feedback. We trust that the explanations provided will satisfactorily address your concerns and contribute to a better understanding of our work.

---

### Official Review · Reviewer_MxEg · 2023-11-06

**Soundness:** 3 good
**Presentation:** 3 good
**Contribution:** 3 good
**Rating:** 8
**Confidence:** 5

**Summary:**

This paper introduces a novel approach to multi-resolution pre-training for speech, designed to leverage information at various resolutions effectively. The authors demonstrate performance improvements in several downstream tasks, such as speech recognition, when compared to the prior work HuBERT.

**Strengths:**

This paper proposes a novel method for pre-training speech data at multiple resolutions within one model. The improvements on several downstream tasks are significant when using unlabelled data at different scales. The figures and tables are also well presented.

**Weaknesses:**

There're two problems need to be solved before acceptance.
1. The positions of $f_1^q$ and $f_2^q$ are reversed in equation 2. According to your description, the $\tilde{H}_0$ is first processed by $f_1^q$.
2. In HuBERT, there's only one output sequence, so it is sent to a CTC layer when fine-tuning. However, in MR-HuBERT, there're two output sequences. How are the two sequences of different lengths combined and sent to CTC? I didn't find the details in the paper.

**Questions:**

See the two problems listed in the above weakness part.

---

> ### Author Response · Authors · 2023-11-11
> **Response to Reviewer MxEg**
>
> Dear Reviewer MxEg,
>
> Thank you for your effort and for your positive feedback on our work.
>
> Regarding your observations:
>
> - We acknowledge the reversal error between $f_1^q$ and $f_2^q$ that you have identified. We will fix this mistake in our forthcoming revised manuscript.
> - Your second point has brought to our attention the need for greater clarity in describing the use cases of pre-trained MR-HuBERT. To address this:
>   - For the fine-tuning setting, we indeed utilize the hidden states from the last layer for downstream models, followed with a shallow linear CTC decoder. It is important to note that the lower-resolution layers are not directly engaged in this process. The intention is to demonstrate the effectiveness of the pre-trained MR-HuBERT as a comprehensive encoder. We will enhance the manuscript by including an additional sentence to elucidate this for the benefit of our readers.
>   - Regarding the frozen setting (SUPERB and ML-SUPERB), we employ a weighted summation of hidden representations from all layers. To synchronize the varying resolutions, we repeat-upsample the time axis of the lower-resolution representations, ensuring alignment across all layers. This approach is in line with established practices found in prior work, which can be reviewed at https://arxiv.org/abs/2306.01084.
>
> We hope that these clarifications satisfactorily address your comments and enhance the understanding of our proposed methodology.

---

> ### Comment · Reviewer_MxEg · 2023-11-11
>
> Thanks for the response. I think the paper can be accepted after the two issues are solved.

---

### Comment · Area_Chair_8xkg · 2023-11-10
**reviewer-author discussions**

Dear All,

The reviewer-author discussion period will be from Nov. 10 to Nov. 22. For reviewers, please read the authors' responses and acknowledge it, respond to them early on in the discussion, and discuss points of disagreement. Thank you!

AC

---

### Author Response · Authors · 2023-11-21
**Paper Revision Summary**

We extend our gratitude to all reviewers for their diligent efforts and valuable input. Based on your feedback, we have made the following revisions to our manuscript:

- Correction of Equation Formulation: We have rectified the incorrect formulation in Equation 2, ensuring the correct order of $f_1^q$ and $f_2^q$.
- Detailed Description of ASR Fine-Tuning Experiments: The manuscript now explicitly states that the entire encoder, specifically the last layer representation, is utilized for the shallow CTC decoder in the ASR fine-tuning experiments (in Section 4.2).
- Inclusion of Missing Variable: We have addressed the omission by including $g^q_{R_2}$ in Equation 5.
- Figure Enhancement: To aid in clearer understanding, we have updated Figure 1 to incorporate corresponding notations that align with the model’s description.
- Additional Information in Appendix C: An Appendix C has been added to provide supplementary details on the real system speed during decoding. This addition offers better reference points for understanding the speed improvements achieved.
- Typographical Corrections: We have corrected several typographical errors as highlighted by Reviewer hKsL.

These amendments aim to enhance the clarity, accuracy, and comprehensiveness of our manuscript, addressing the concerns and suggestions raised from the reviewers. We believe these changes substantially improve the quality of our submission. We are happy to clarify any remaining points (if any).

Thanks in advance,
Submission 2381 authors

---

### Meta-Review · Area_Chair_8xkg · 2023-12-06

**Metareview:**

This paper proposes a novel method for self-supervised speech representation learning that leverages multi-resolution information from speech signals. The authors introduce a hierarchical Transformer architecture that processes speech at two resolutions and applies HuBERT-style masked prediction objectives at both levels.

This is the first work to explicitly address the integration of multi-resolution information into the pre-training of a single model for speech.

The proposed method has been verified on several benchmarks with extensive experiments. The most notable results are in the SUPERB benchmark which contains multiple speech tasks, the proposed method usually performs better than HUBERT. This clearly demonstrated such multi-resolution strategy can be beneficial to the SSL learning as a general representation for all kinds of speech tasks.

In addition to these major contributions, the proposed method exhibits computational efficiency. Therefore, this paper should be a "defintely accept" paper which brings impacts to the speech community.

This paper still has some weaknesses:

The improvement on some tasks may not be significant. (98.76 vs. 98.73 in IC, 97.76 vs. 97.70 in KS, 2.67 vs. 2.65 in SE-PSEQ).

The HUBERT baseline is not the best in SUPERB. Why not use the best model wavLM to verify the proposed multi-resolution method? Then, the proposed method can claim the best in the SUPERB leaderboard and will be more convincing.

The performance of the proposed MR-HUBERT on ASR task in SUPERB is not good compared to HUBERT, although the authors argue that the ASR in SUPERB is not stable.

**Justification For Why Not Higher Score:**

The paper still has some weakness, especially this multi-resolution model was built on top HUBERT instead of wavLM which is the best model in the SUPERB benchmark. If the paper can demonstrate its effectiveness on top of wavLM, then it deserves the highest score.

**Justification For Why Not Lower Score:**

This paper has lots of contributions to the speech community, and has enough novelty. The experiments are performed extensively to verify its effectivenss, and the results on SUPERB are espeically convincing.

---

### Decision · Program_Chairs · 2024-01-16

Accept (spotlight)